# GRAPH NEURAL NETWORKS GONE HOGWILD

## ABSTRACT

Graph neural networks (GNNs) constitute a dominant class of architectures for modeling graph-structured data. Message-passing GNNs in particular appear to be ideal for applications where distributed inference is desired, since node updates can be performed locally. In this work, we are particularly motivated by the view that GNNs can be interpreted as parametric communication policies between agents which collectively solve a distributed optimization problem (e.g., in robotic swarms or sensor networks). For these applications, node synchrony and central control are undesirable, since they result in communication bottlenecks and reduce fault tolerance and scalability. We examine GNN inference under asynchrony, and find that most GNNs generate arbitrarily incorrect predictions in this regime. A notable exception is GNNs which cast message passing as a fixed point iteration with contractive update functions. We propose a novel GNN architecture, *energy GNN*, in which node embeddings are computed by minimizing a scalar-valued convex function which we call an 'energy' function. By framing message passing as convex optimization, we unlock a richer class of update functions which preserve robustness under asynchronous execution. We show that, empirically, we outperform other GNNs which are amenable to asynchronous execution on a multitude of tasks across both synthetic and real-world datasets.

## 1 INTRODUCTION

Graph neural networks (GNNs) have gained prominence as a powerful framework for deep learning on graph-structured data, finding success in application domains like molecular chemistry (Duvenaud et al., 2015), social networks, and recommendation systems (Fan et al., 2019). GNNs use message passing within local graph neighborhoods to effectively produce a deep neural network architecture whose computational graph reflects the structure of the input graph. Neural network architectures exhibiting equivariance and/or invariance have been critical to the success of deep learning, and GNNs can be viewed as a way to generalize these concepts to graph-structured data.

At first glance, the message passing framework appears to be a prime candidate for distributed and decentralized execution, which is desirable in a variety of contexts. Consider a group of agents (e.g., robots or "motes" in a sensor network) which need to collectively perform a task, but that might be unreliable, have limited range of communication, possess scarce local computational resources, and lack central control. GNNs are appealing as a way to learn local *communication policies* that solve the distributed problem, where each agent corresponds to a node in the graph and the edges correspond to local communication constraints. One could imagine learning an algorithm that enables a swarm of robots to localize themselves, or a collection of resource-constrained edge devices to collectively estimate environmental conditions. Another application where distributed computation is attractive is in GNN inference over large graphs, where nodes or node collections are managed on distinct machines (respecting the graph connectivity). This is especially relevant for GNN deployment on resource-constrained devices. Distributed inference may also facilitate privacy in settings where nodes correspond to entities such as individuals in a social network, by enabling local inference and precluding excessive data transmission.

There is a significant defect in this analogy: distributed and decentralized computation is generally asynchronous, and existing GNN architectures implicitly assume synchronism at inference time. That is, the parameters of the GNN are trained such that the correct computation is performed provided there are synchronous rounds of message passing per layer. When nodes update at different times or messages are stale, the effective architecture diverges catastrophically from the training

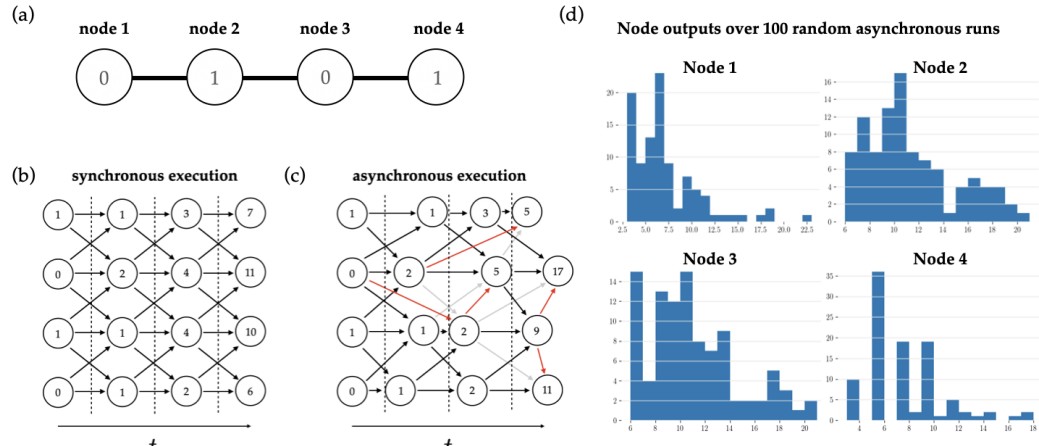

Figure 1: Modifications to computation graph of an $L$-layer message-passing GNN resulting from asynchronous, distributed, per-node inference. **(a)** An undirected linear graph with binary features written in the node body. **(b)** Synchronous per-layer execution of a 3-layer GNN on the graph from (a); all nodes update at the same time, using neighbor information from the previous layer. Arrows represent network weights. For demonstration, weights are all equal to 1, and the value of a node embedding at the next layer is the sum of incoming values. **(c)** Asynchronous inference of the GNN from (b). Nodes update at random times and can use neighbor information corresponding to the incorrect layer, which introduces modifications to the computation graph. Gray arrows correspond to connections that were removed from the original computation graph, and red arrows are unintended connections resulting from asynchrony. **(d)** To demonstrate the effect of asynchrony, we show the output of the GNN varies significantly over asynchronous runs with different node update orderings.

architecture; this means the output of the GNN can exhibit arbitrarily large errors. Figure 1 illustrates the issue on a chain graph.

We observe that certain classes of GNNs, which might be termed "hogwild-able" GNNs (inspired by Recht et al. (2011), are *provably robust* to asynchrony given lenient assumptions on staleness and per-node update frequency. For instance, implicit GNNs Gu et al. (2020); Liu et al. (2021), which use a fixed point iteration to implement message passing, are 'hogwild-able'. We introduce an alternative to achieve robustness to asynchrony: reframing message passing as an optimization procedure over a convex global graph function. We propose a novel hogwild-able GNN architecture, which we refer to as an *energy GNN*. In an energy GNN, node embeddings are computed as the minimum of a convex function which is implemented via input-convex neural networks defined on nodal neighborhoods. We loosely interpret the function being minimized as an 'energy', as an analogy to physical systems which naturally tend to minimize their energy to achieve stable configurations. We show that energy GNN outperforms other hogwild-able GNNs in a variety of synthetic tasks, particularly those which demand long-range communication between nodes or the use of edge features for compelling performance. We also achieve competitive performance on benchmark datasets, certifying the merit of our approach even as a stand-alone GNN architecture.

Section 3 provides a brief review of message passing GNNs. In section 4, we introduce our framework for asynchronous and distributed inference in GNNs. In section 5 we present our architecture: energy GNNs. Section 6 presents experimental results, where we evaluate performance during inference using both synchronous and asynchronous forward execution.

## 2 RELATED WORK

### 2.1 DISTRIBUTED OPTIMIZATION AND ASYNCHRONOUS ALGORITHMS

Asynchronous algorithmic models (sometimes called *chaotic relaxation* models) date back at least to the late 1960s (Chazan & Miranker, 1969), and were explored extensively into the 1970s and 1980s (Donnelly, 1971; Miellou, 1975; Robert et al., 1975; Baudet, 1978; Bertsekas, 1982; 1983; Bojańczyk, 1984; Mitra, 1987; Uresin & Dubois, 1989). In appendix A.1 we study GNNs under *partial asynchronism*, a particular model which imposes constraints on the sequencing of computations and the frequency of communication between distributed elements. Our analysis of partially asyn-

chronous GNN inference draws directly from prior work which analyzes the sufficient conditions for convergence (Tsitsiklis, 1984; Tsitsiklis et al., 1986; Bertsekas & Tsitsiklis, 1989).

Interest in distributed computing and optimization originated at a similar time, coincident with the unprecendented rate of progress in digital electronics and computer engineering (Borodin & Munro, 1975; Goldschlager, 1978; Hockney & Jesshope, 1981; Hwang, 1984; Quinn, 1987). The project of scaling optimization and machine learning systems to extremely large datasets has sustained interest in decentralized, distributed computation (Bottou et al., 2018; Yang et al., 2019). Like energy GNNs, many of the problems can be formulated in the framework of convex optimization, in fact distributed convex optimization is an area of interest in its own right (Boyd et al., 2011).

## 2.2 ASYNCHRONICITY IN GRAPH NEURAL NETWORKS

There has been a plethora of work in the area of distributed *training* of GNNs, where the data is partitioned and computed on by separate workers (Besta & Hoefler, 2022; Shao et al., 2022). In this training regime, workers may not be operating on independent data, e.g., when data partitions come from a single connected graph. Since ignoring this dependence reduces application performance, workers exchange embedding information associated with "boundary vertices" which are logically connected but delegated to different workers. Some distributed training frameworks assume workers operate asynchronously, using stale embedding (Md et al., 2021; Peng et al., 2022; Wan et al., 2022) or gradient information (Thorpe et al., 2021) (corresponding to previous training epochs) from other workers. However, across all these frameworks, the forward pass is executed synchronously per layer, and to our knowledge there is no work examining asynchronous GNN inference.

## 2.3 IMPLICIT GNNS

Like energy GNNs, implicit GNNs Scarselli et al. (2009); Gu et al. (2020); Liu et al. (2021) define node embeddings implicitly as the solution to an iterative algorithm. Specifically, these architectures obtain node embeddings via a fixed point iteration on a contractive node embedding update function. Since the number of iterations is not predetermined, implicit GNNs are sometimes referred to as "infinite-depth" GNNs. Implicit GNNs which use contractive node updates are extremely well suited for partially asynchronous, decentralized, and distributed inference. This is because under reasonable assumptions (see appendix A.1), it follows from the contractive property of the node updates that the embeddings converge (Bertsekas, 1983; Bertsekas & Tsitsiklis, 1989).

That said, existing implicit architectures use relatively simple updates which can easily verified and enforced to be contractive. The original 'nonlinear GNN' proposed by Scarselli et al. (2009) is an exception, but this comes at a cost, as their method encourages rather than guarantees contraction. Their strategy is to formulate a multi-objective problem in which the norm of the Jacobian of the update function at the fixed point is also to be minimized. This heuristic can work in practice, but the sequence of iterates does not definitively converge, particularly if node embeddings are initializd far from the fixed point solution (as the norm of the Jacobian is only penalized at the fixed point).

Another difficulty with implicit GNNs is that both the forward and backward passes of the network are usually implemented with iterative solvers. The number of iterations required to converge is not known in advance, and training is sensitive to hyper-parameter choices for the solvers. To alleviate this, EIGNN Liu et al. (2021) derive a contractive fixed point update whose limit can be computed efficiently in closed form. This closed form solution requires global information and is not amenable to distributed inference. Interestingly, we observe that at inference time, results from iterative execution differ significantly from those achieved by the closed form (see appendix A.10).

## 3 GRAPH NEURAL NETWORKS

We provide a brief overview of GNNs, loosely following notation used in (Hamilton, 2020). Consider a directed graph with $n$ vertices $V = \{1, ..., n\}$, and edges $E \subseteq V \times V$. The connectivity of the graph is contained in its adjacency matrix $\boldsymbol{A} \in \{0, 1\}^{n \times n}$, where $\boldsymbol{A}_{i,j} = 1$ if there is an edge from node $i$ to node $j$, and $0$ otherwise. The graph may also have associated node and edge features $\boldsymbol{X} \in \mathbb{R}^{n \times p}$ and $\boldsymbol{E} \in \mathbb{R}^{|E| \times q}$, so we use $\mathcal{G} = (\boldsymbol{A}, \boldsymbol{X}, \boldsymbol{E})$ to denote the graph (and its features).

Two canonical prediction tasks are classification and regression at the graph or node level, in which we want a vector representation either of the graph or of each node which is useful in the task.

We focus on node-level embeddings, since they often underly graph-level embeddings. We are given a dataset $\mathcal{D} = \{(\mathcal{G}, \boldsymbol{Y})^d\}_{d=1}^{|\mathcal{D}|}$, where each $\mathcal{G}^d$ is associated with a node-level prediction target $\boldsymbol{Y}^d \in \mathbb{R}^{n_d \times \ell}$, where $n_d$ is the number of nodes in graph $d$.

In their most general form, GNNs define a parameterized embedding function $f_\theta : \mathcal{G} \to \mathbb{R}^{n \times k}$ which takes as input the graph data and parameters and returns a k-vector embedding $\boldsymbol{h}_i$ for each node. A readout function (often a linear transformation) $o_\phi : \mathbb{R}^k \to \mathbb{R}^\ell$ is applied to each embedding, which results in node predictions $\hat{\boldsymbol{Y}} = (o_\phi(\boldsymbol{h}_1), ..., o_\phi(\boldsymbol{h}_n))^T \in \mathbb{R}^{n \times \ell}$. Given a task-specific loss $\mathcal{L}$, training of the GNN corresponds to the following optimization problem:

$$\theta, \phi = \underset{\theta, \phi}{\arg\min} \frac{1}{|\mathcal{D}|} \sum_{d=1}^{|\mathcal{D}|} \mathcal{L}(\hat{\boldsymbol{Y}}^d, \boldsymbol{Y}^d). \tag{1}$$

### 3.1 MESSAGE PASSING GNNS

Most GNNs use message passing in the embedding function $f_\theta$. At each iteration of message passing, each node $i$ receives messages $\boldsymbol{m}_{ij}$ from nodes $j$ in its local neighborhood. Each $\boldsymbol{m}_{ij}$ is obtained by applying a function $m$ to information pertaining to that neighbor relation (e.g., embeddings and node/edge features). The messages are aggregated into a single message $\boldsymbol{m}_i$ via a permutation-invariant aggregation function $g$. Finally, an update function $u$ uses a node's aggregated message to update its embedding. We use $\theta_m, \theta_g, \theta_u$ to denote the subsets of the parameters $\theta$ used in each of $m, g, u$, respectively. GNNs often consist of several iterations (or "layers") of message passing; node embeddings are updated $L$ times and each iteration may have distinct functional forms and parameters for $m, g,$ and $u$. A node $i$'s embedding at iteration $\ell \in \{0, ..., L\}$ is denoted by $\boldsymbol{h}_i^\ell \in \mathbb{R}^{k^{(\ell)}}$, and the final layer embeddings $\boldsymbol{h}_i^L$ are used as input to the readout function $o_\phi$. In its most general form, the embedding update function at iteration $\ell$, $f_\theta^\ell$, can be written as:

$$\boldsymbol{m}_{ij}^\ell := m^\ell(\boldsymbol{h}_j^\ell, \boldsymbol{h}_i^\ell, \boldsymbol{X}_j, \boldsymbol{X}_i, \boldsymbol{E}_{ij}; \theta_m^l) \quad \forall i, j \in E \qquad \text{(Create message on edge } i, j) \tag{2}$$

$$\boldsymbol{m}_i^\ell := g^\ell\left(\{\boldsymbol{m}_{ij}^\ell \mid j \in \text{ne}(i)\}, \boldsymbol{A}_i; \theta_g^\ell\right) \qquad \text{(Aggregate message on node } i) \tag{3}$$

$$\boldsymbol{h}_i^{\ell+1} := u^\ell\left(\boldsymbol{m}_i^\ell, \boldsymbol{h}_i^\ell, \boldsymbol{X}_i; \theta_u^\ell\right), \qquad \text{(Update hidden state of node } i) \tag{4}$$

where $\text{ne}(i)$ denotes the neighbors of node $i$ and $\boldsymbol{E}_{ij}$ are the edge features for the edge from $j$ to $i$. Many GNNs initialize the node embeddings at $\ell = 0$ to be equal to the node features (or some simple function of the node features), and use the entries of the adjacency matrix $\boldsymbol{A}$ (or some variant of the adjacency matrix) in $g$. The particular form of $m, g,$ and $u$ varies across GNNs; below we describe several concrete examples which we reference throughout the paper. We use $\boldsymbol{H}^\ell \in \mathbb{R}^{n \times k^{(\ell)}}$ to denote the all of the node embeddings at iteration $\ell$.

**Graph Convolutional Networks** GCNs (Kipf & Welling, 2017) define $g$ as a weighted sum of neighbor messages based on the entries of the symmetric normalized adjacency matrix with added self-loops, $\tilde{\boldsymbol{A}} = (\boldsymbol{D} + \boldsymbol{I})^{-\frac{1}{2}}(\boldsymbol{A} + \boldsymbol{I})(\boldsymbol{D} + \boldsymbol{I})^{-\frac{1}{2}}$. With node embeddings initialized to be equal to the node features, $f_\theta^\ell$ is defined as:

$$\boldsymbol{m}_i^\ell := \sum_{j \in \text{ne}(i)} \tilde{\boldsymbol{A}}_{i,j} \theta_m^\ell \boldsymbol{h}_j^\ell \qquad\qquad \boldsymbol{h}_i^{\ell+1} := \text{ReLU}(\boldsymbol{m}_i^\ell), \tag{5}$$

where $\theta_m^\ell \in \mathbb{R}^{k^{(\ell)} \times k^{(\ell)}}$. This update can be succinctly described at the graph level as $\boldsymbol{H}^{\ell+1} = \text{ReLU}(\tilde{\boldsymbol{A}} \boldsymbol{H}^\ell \boldsymbol{W}^\ell)$. Note that for finite depth message passing GNNs which have $L$ layers, such as GCN, it is impossible to propagate information farther than $L$ hops.

**Implicit Graph Neural Networks** Implicit GNNs (Scarselli et al., 2009; Gu et al., 2020; Liu et al., 2021) are "infinite-depth" GNNs, where the number of iterations of message passing is not predetermined, and instead a single parameterized layer is repeated as many times as is required to numerically reach a fixed point. The IGNN architecture Gu et al. (2020) uses a similar embedding update function as GCN, but adds node features as an additional input to $u$. A layer is defined at the graph level as:

$$\boldsymbol{H}^{\ell+1} := \phi(\tilde{\boldsymbol{A}} \boldsymbol{H}^\ell \theta_m + \boldsymbol{X} \theta_u), \tag{6}$$

where $\theta_m, \theta_u \in \mathbb{R}^{k \times k}$ and $\phi$ is a component-wise non-expansive function such as ReLU. Convergence is guaranteed by constraining $||\theta_m||_\infty < \lambda_{max}(\tilde{\boldsymbol{A}})^{-1}$, where $\lambda_{max}$ is the maximum eigenvalue of $\tilde{\boldsymbol{A}}$. This ensures that the update is contractive, a sufficient condition for convergence.

## 4 ASYNCHRONOUS AND DISTRIBUTED INFERENCE IN GNNS

GNNs assume that layer updates are performed synchronously, as depicted in Figure 1, where each node embedding is updated using the previous layer embeddings. As discussed previously, we identify two flavors of problems in which asynchronous, distributed execution is desirable, requiring us to break the synchronicity assumption of GNNs. The first is using GNNs to parameterize communication protocols between simple computational agents which have limited computational resources and range of communication, and the second is distributed execution of GNNs on large graphs. The existence of relevant applications motivates an analysis of asynchronous, distributed GNN inference. To our knowledge, this inference regime has not been explored, so we first describe partially asynchronous algorithms, and then outline GNN inference under partial asynchrony. We focus on per-node inference for clarity, but in practice each node can correspond to a worker operating on a graph partition rather than a single node.

Computational models for asynchronous algorithms vary depending on the constraints imposed on the sequencing or frequency of computation or communication. We consider *partial asynchronism*, which, informally, places bounds on two key characteristics: the time between updates across each node, and the amount by which data retained at any node can be out of date (from the perspective of some other node(s)). In this section we present GNN inference under partial asynchronism. We give a brief but more precise overview of the algorithmic model in appendix A.1, see (Bertsekas & Tsitsiklis, 1989) for a thorough coverage.

We write $\boldsymbol{h} = (\boldsymbol{h}_1, \boldsymbol{h}_2, \ldots, \boldsymbol{h}_n) \in \mathbb{R}^d$ to denote a block vector containing the embedding data $\boldsymbol{h}_i \in \mathbb{R}^k$ associated with each of the $n$ nodes. We define a collection of local or 'node-specific' update functions $f_i : \mathbb{R}^d \mapsto \mathbb{R}^k$, which are essentially embedding updates $f_\theta$ restricted to node neighborhoods. Without loss of generality, assume the update functions are continuously differentiable, so that this restriction can be stated:

$$j \notin \text{ne}(i) \implies \tfrac{\partial f_i}{\partial \boldsymbol{h}_j}(\boldsymbol{z}) = 0 \quad \forall \boldsymbol{z} \in \mathbb{R}^k. \tag{7}$$

For inference under asynchronism, we aim to coordinate these nodes, so that in iterating the local updates $f_i$ using local neighborhood data, the sequence of embeddings (across the graph) converges. We must reason about particular orderings of the local node updates, so we consider the embeddings as a function of time.

Suppose we are given a set $T^i \subseteq \{0, 1, 2, \ldots\}$ of times at which the node $i$ is updated. For each $t \in T^i$ we are given variables $\tau_j^i(t), \quad i, j = 1, \ldots, n$. The latter satisfy $0 \leq \tau_j^i(t) \leq t$, and can be interpreted as the time $\tau_j^i(t) \in T^j$ corresponding to node $i$'s view of node $j$ at time $t$. The quantities $s_{ij}(t) = t - \tau_j^i(t) \in [0, t]$ can be interpreted as the amount (in time) by which information associated with node $j$ is outdated or "stale" when used in the update of $\boldsymbol{h}_i$ at time $t$.

For simplicity, assume that the embedding dimension $k$ is fixed, so embeddings for a node are compatible with whatever update iteration the node is at. Additionally, we assume that the number of iterations $|T_i|$ executed by each node is fixed and equal to the number of layers $L$.

For correspondence with the eq. (3) and eq. (4), we write the update for a *single* layer:

$$\boldsymbol{m}_{ij}(t+1) \coloneqq m(\boldsymbol{h}_j(\tau_j^i(t)), \boldsymbol{h}_i(t), \boldsymbol{X}_j, \boldsymbol{X}_i, \boldsymbol{E}_{ij}; \theta_m) \tag{8}$$

$$\boldsymbol{m}_i(t+1) \coloneqq g\left(\{\boldsymbol{m}_{ij}(t+1) \mid j \in \text{ne}(i)\}, \boldsymbol{A}_i; \theta_g\right) \tag{9}$$

$$\boldsymbol{h}_i(t+1) \coloneqq u\left(\boldsymbol{m}_i(t+1), \boldsymbol{h}_i(t), \boldsymbol{X}_i; \theta_u\right), \tag{10}$$

for $t \geq 0$. Note the crucial difference introduced by partial asynchronism: in computing $\boldsymbol{m}_i$ (the message associated with node $i$), the neighbor data $\boldsymbol{h}_j(\tau_j^i(t))$ may be out of date. As written, this update corresponds to node $i$ executing a single layer. For simplicity, we do not write the update functions $m, g, u$ or their parameters $\theta_m, \theta_g, \theta_u$ indexed by layer, but this is straightforwardly generalized to the case of layer-specific parameters and functions described in eqs. (2) to (4).

In our experiments, partially asynchronous execution is simulated; we give details of our implementation in appendix A.5. In the framework laid out here, implicit GNNs enjoy two advantages compared to finite depth GNNs. First, provided the fixed point iteration is contractive, the embeddings converge under partial asynchronism (Bertsekas, 1983). In contrast, finite depth GNNs implement a specific feedforward neural network architecture, and partial asynchrony corrupts the

computation performed by the network. We illustrate this in Figure 1, where partial asynchrony results in a (different) computation graph with some connections removed, and new connections that are not present in the original synchronous computation graph. In general, this means the final node embeddings may vary significantly depending on the particular node update sequence.

Second, implicit GNNs can continue to iterate indefinitely, so by construction they are adaptive to dynamic inputs. Put another way, if node or edge features are *not* time-invariant, an iterating implicit GNN will eventually change its output in response to changes in the inputs. On the other hand, in finite depth GNNs each node is constrained to execute exactly $L$ updates, and there is no straightforward solution to the problem of coordinating another forward pass of the network.

## 5 ENERGY GRAPH NEURAL NETWORKS

As discussed in the previous section, implicit GNNs which frame message-passing as a fixed point iteration are perfectly suited for decentralized, distributed, and asyncronous inference. However, existing implicit GNNs either (1) use simple update functions that can easily be enforced to be contractive Gu et al. (2020); Liu et al. (2021), or (2) attempt to specify more flexible update functions which are encouraged (rather than guaranteed) to be contractive. An alternative strategy to constructing implicit GNNs is to replace the fixed point iteration with an optimization procedure. In other words, the GNN embedding updates can be viewed as iterations in an algorithm for solving an optimization problem. Previous work has explored the relationship between existing GNN updates and optimization objectives, and propose generalized forms of objectives which unify the optimization-oriented view of GNNs (Yang et al., 2021, Zhu et al., 2021). As an example, we derive the optimization objective associated with EIGNNs Liu et al. (2021) in Appendix A.9. However, there is no reason to limit the design space of GNN updates to correspond to the functional form of the objective proposed in previous work.

We propose a novel implicit GNN architecture which we call *energy GNNs*, motivated by the optimization-oriented view of GNN updates. Energy GNNs compute node embeddings that minimize a parameterized, convex function, which we refer to as the 'energy' function. As we discuss later, this formulation enables robustness to distributed and partially asynchronous inference, like other implicit GNNs which use contractive fixed point node updates. By employing partially input-convex neural networks (PICNNs) in the architecture of the energy function, we open a rich, flexible class of convex graph objectives.

### 5.1 INPUT-CONVEX GRAPH NEURAL NETWORKS

PICNNs (Amos et al., 2017) are scalar-valued neural networks that constrain the parameters in such a way that the network is convex with respect to a subset of the inputs. We use a partially input-convex GNN (PICGNN) as the energy function by extending PICNNs to operate on graph-structured data. A regular message passing GNN can be recast to be convex with respect to the node embeddings with two modifications. First, we use PICNNs for the functions $m$, $u$, and $o_\phi$, where the functions are convex with respect to the messages $\boldsymbol{m}$ and node embeddings $\boldsymbol{H}$, but not necessarily with respect to the features. Second, non-negative summation is used for the aggregation functions (which preserves convexity). We provide a more detailed description of the architecture of a general PICGNN in Appendix A.2.

### 5.2 ENERGY FORMULATION

An energy GNN replaces the GNN embedding update function $f_\theta$ with an optimization procedure which minimizes an energy function $E_\theta$ with respect to the node embeddings. We use a PICGNN as $E_\theta$ and define $\theta = (\bar{\theta}, \bar{\phi})$ to be the parameters of the energy function, and $(\theta, \phi)$ to be the parameters of the energy GNN. In our experiments, $E_\theta$ is the sum of the node-level outputs $e_i \in \mathbb{R}$ which in turn are the sum of the layer output and the squared $L^2$ norm of the node embeddings:

$$E_\theta(\mathcal{G}, \boldsymbol{H}) = \sum_{i=1}^{n} e_i \qquad \text{where} \tag{11}$$

$$\boldsymbol{m}_i = \sum_{j \in \text{ne}(i)} \boldsymbol{A}_{i,j} m(\boldsymbol{h}_j, \boldsymbol{h}_i, \boldsymbol{X}_j, \boldsymbol{X}_i, \boldsymbol{E}_{ij}; \bar{\theta}_m) \qquad e_i = u\left(\boldsymbol{m}_i, \boldsymbol{h}_i, \boldsymbol{X}_i; \bar{\theta}_u\right) + (\beta/2)||\boldsymbol{h}_i||_2^2. \tag{12}$$

In the forward pass, node embeddings are obtained by minimizing $E_\theta$ with respect to $\boldsymbol{H}$:

$$\boldsymbol{H}^* = \arg\min_{\boldsymbol{H}} E_\theta(\mathcal{G}; \boldsymbol{H}). \tag{13}$$

Node-level predictions are then obtained using a neural network output function $o_\phi$ which takes as input the energy-minimizing embeddings. We use gradient descent to solve for $\boldsymbol{H}^*$, although in principle any convex optimization procedure can be used. More formally, we initialize $\boldsymbol{H}^0 = \boldsymbol{0}$ and for iterations $t = 0, ..., T-1$ perform the following update:

$$\boldsymbol{H}^{t+1} = \boldsymbol{H}^t - \alpha \frac{\partial E_\theta}{\partial \boldsymbol{H}}(\boldsymbol{H}^t), \tag{14}$$

where $\alpha > 0$ is the step size and the number of iterations $T$ is dictated by when the embeddings numerically reach a fixed point $\boldsymbol{H}^*$; i.e., when $\boldsymbol{H}^{t+1} \approx \boldsymbol{H}^t$. The node-level view of Equation 14 makes it clear that just like in regular message passing GNNs, updates are performed per node using information from directly connected neighbors:

$$\boldsymbol{h}_i^{t+1} = \boldsymbol{h}_i^t - \alpha \sum_{j \in \text{ne}(i) \cup \{i\}} \frac{\partial e_i}{\partial \boldsymbol{h}_i}(\{\boldsymbol{h}_{j'}^t | j' \in \text{ne}(j)\}), \tag{15}$$

We prove in Appendix A.4 that since the $E_\theta$ is strongly convex and separable per node, this optimization procedure converges under partial asynchronism and can be executed in a distributed manner.

We exploit convergence of the energy minimization process by using implicit differentiation to obtain gradients of the task-specific loss function $\mathcal{L}$ with respect to the energy parameters $\theta$. This avoids unrolling the iterations of the energy minimization procedure in the backward pass, and requires a fixed amount of computation and memory. We derive the gradient and provide additional details in Appendix A.3.

### 5.3 PARTIALLY ASYNCHRONOUS INFERENCE

In order to examine energy GNN inference under partial asynchronism, we associate each node $i$ with a $k$-vector embedding $\boldsymbol{h}_i \in \mathbb{R}^k$, and a collection of $m_i = |\text{ne}(i)|$ additional $k$-vectors $\boldsymbol{g}_{ij} \in \mathbb{R}^k$, $j = 1, \ldots, m_i$, one for each of the $m_i$ neighbors of node $i$. The meaning of $\boldsymbol{g}_{ij}$ is the derivative of $e_i$ with respect to the (potentially outdated) node embedding $\boldsymbol{h}_j$ associated with node $j$. We collect all the data associated with node $i$ into a length $d_i = k(m_i + 1)$ block vector $\boldsymbol{x}_i = (\boldsymbol{h}_i, \boldsymbol{g}_{i1}, \ldots, \boldsymbol{g}_{im}) \in \mathbb{R}^{d_i}$. We then aggregate the data associated with all $n$ nodes into a block vector with $d = \sum_{i=1}^n d_i$ elements which we denote $\boldsymbol{x} = (\boldsymbol{x}_1, \boldsymbol{x}_2, \ldots, \boldsymbol{x}_n) \in \mathbb{R}^d$; this corresponds with the state variables $\boldsymbol{x}$ defined in appendix A.1.

The node data evolves according to update functions $f_i : \mathbb{R}^d \mapsto \mathbb{R}^{d_i}$, $i = 1, \ldots, n$. Each update function $f_i$ is comprised of (1) a gradient descent step on the energy function with respect to $\boldsymbol{h}_i$ and (2) derivative estimates $\tilde{\boldsymbol{g}}_{ij} = \frac{\partial e_i}{\partial \boldsymbol{h}_j}(\boldsymbol{h}_j(\tau_j^i(t))$ given (potentially updated) embeddings $\boldsymbol{h}_j(\tau_j^i(t))$ associated with the node's neighbors.

## 6 EXPERIMENTS

We perform experiments on a number of synthetic datasets, motivated by tasks which are of interest for multi-agent systems and where distributed, asynchronous inference is desirable. Specifically, we examine the ability of GNNs to capture long-range dependencies between nodes, perform size estimation and summation, and perform relative node localization. We compare performance of energy GNNs to IGNN Gu et al. (2020) and GCN Kipf & Welling (2017). IGNN is compared against because it is the main existing GNN architecture that we identify to be amenable to asynchronous inference. Two other architectures mentioned in section 2.3 that are excluded are EIGNN Liu et al. (2021) and the GNN proposed by Scarselli et al. (2009). The former is excluded because the fixed point is solved for directly in the forward pass rather than iteratively, which requires global information, and the latter is excluded because fixed point convergence is encouraged rather than guaranteed. GCN is chosen as a representative architecture from the class of finite depth GNNs and we do not consider other architectures, since all GNNs from this class exhibit the same malignancies under asynchronous inference and are thus not compatible with the synthetic tasks. Performance of GCN under synchronous and asynchronous inference is provided as a reference, as well as to further demonstrate the deviation in predictions under asynchrony. We use 2 layers of message passing for GCN, and for IGNN and energy GNN we use a single parameterized layer of message passing.

For all synthetic experiments, we simulate asynchronous inference of trained models. For IGNN and energy GNN, we report results under this regime, since both achieve (within numerical error) the same performance under synchrony or asynchrony. For GCNs, we report results with both synchronous and asynchronous inference as they deviate significantly. Since the output from asynchronous execution of a GCN depends on the order of node updates, we report mean performance across 10 random orderings. The simulated asynchronous inference algorithm is in Appendix A.5. Training details for the synthetic experiments are provided in Appendix A.6.

We additionally perform experiments on benchmark datasets (MUTAG Srinivasan et al. (1996), PROTEINS Borgwardt et al. (2005), PPI Hamilton et al. (2017)) for node and graph classification to evaluate energy GNNs as a synchronous GNN architecture, and achieve competitive performance on each dataset. Details related to these experiments are provided in Appendix A.8.

## 6.1 CHAINS

In the absence of a central controller (as is the case for distributed, asyncronous inference), the ability of a GNN to capture long-range dependencies between node embeddings depends entirely on local message passing. The chains dataset, used in Gu et al. (2020); Liu et al. (2021), is meant to evaluate this ability. The dataset consists of $p$ undirected linear graphs with $l$ nodes, with each graph having a label $k \in \{1, ..., p\}$. The task is node classification of the graph label, where class information is contained only in the feature of the first node in the chain; the node feature matrix $\boldsymbol{X} \in \mathbb{R}^{n \times p}$ for a graph with class $k$ has $\boldsymbol{X}_{1,k} = 1$ and zeros at all other indices. Perfect classification accuracy indicates that information is successfully propagated to the final node in the chain.

Table 1 shows binary classification accuracy for chains of lengths $l = \{10, 20, 50, 100\}$. Both energy GNNs and IGNNs achieve perfect accuracy up to 50 nodes, with performance declining slightly at 100 nodes for IGNN. In Appendix A.7,

Table 1: Node classification accuracy (%) for chains dataset, mean and standard deviation across 3 random parameter seeds.

| | **# NODES** | | | |
| MODEL | **10** | **20** | **50** | **100** |
|---|---|---|---|---|
| GCN (sync) | $65.0 \pm 0.0$ | $57.5 \pm 0.0$ | $53.0 \pm 0.0$ | $51.5 \pm 0.0$ |
| GCN (async) | $62.3 \pm 2.9$ | $56.9 \pm 1.7$ | $52.3 \pm 0.2$ | $50.9 \pm 0.1$ |
| IGNN | $100.0 \pm 0.0$ | $100.0 \pm 0.0$ | $100.0 \pm 0.0$ | $93.3 \pm 1.1$ |
| Energy GNN | $\mathbf{100.0 \pm 0.0}$ | $\mathbf{100.0 \pm 0.0}$ | $\mathbf{100.0 \pm 0.0}$ | $\mathbf{100.0 \pm 0.0}$ |

we show plots of dataset loss convergence over the course of asynchronous inference, demonstrating convergence of energy GNN and IGNN predictions.

## 6.2 SUMS

We construct a synthetic dataset meant to test the ability of a GNN to implement a simple distributed function: summation. We consider two regression experiments, node counting and node feature summation in undirected chain graphs.

Table 2: Relative dataset RMSE (%) for counting and summing experiments for chain graphs, mean and standard deviation across 10 folds and 3 random parameter seeds.

| | **EXPERIMENT** | | | |
| MODEL | **COUNT (10)** | **COUNT (50)** | **SUM (10)** | **SUM (50)** |
|---|---|---|---|---|
| GCN (sync) | $61.8 \pm 74.9$ | $41.7 \pm 7.0$ | $24.3 \pm 4.4$ | $14.3 \pm 2.7$ |
| GCN (async) | $199.6 \pm 55.9$ | $108.9 \pm 35.9$ | $47.1 \pm 12.2$ | $14.8 \pm 2.8$ |
| IGNN | $3.7 \pm 1.9$ | $26.1 \pm 3.6$ | $14.7 \pm 2.9$ | $13.0 \pm 2.1$ |
| Energy GNN | $\mathbf{2.9 \pm 1.4}$ | $\mathbf{11.5 \pm 4.5}$ | $\mathbf{2.9 \pm 1.2}$ | $\mathbf{9.9 \pm 1.6}$ |

The dataset for node counting consists of graphs with different numbers of nodes, and no node features. For GCNs and IGNNs, which require node features as input, we use one-hot embeddings of node degrees. We consider two dataset sizes, with ranges of 1-10 and 1-50 nodes, respectively. The prediction target for each node is the total number of nodes in the graphs.

The dataset for node feature summation consists of graphs of the same size, with different instantiations of binary node features $\boldsymbol{X}_i \in \{0, 1\}$. We consider two datasets using 100 graphs with 10 and 50 nodes, respectively. The prediction target for each node is the sum of the graph node features.

Table 2 shows that energy GNN achieves the best relative test RMSE for each dataset.

### 6.3 COORDINATES

A common task for multi-agent collectives such as robot swarms is localization. This problem has previously been tackled in various ways that all employ a bespoke algorithm tailored for the task Todescato et al. (2016); Huang & Tian (2017; 2018). We test the ability of of-the-shelf GNNs to solve this problem on static graphs. We construct a dataset where each node has a position in $\mathbb{R}^2$ and neighbors within some radius are connected by an edge. We don't assume a global coordinate system; instead, we focus on relative localization, where pairwise distances between nodes are maintained. Each node predicts a position in $\mathbb{R}^2$, and the objective is the mean squared error between true pairwise node distances, and distances between their predicted positions. In order to break symmetries, each node has a unique ID which is one-hot encoded and used as the node feature. Distances to connected neighbors are provided as edge features.

We consider two types of datasets; one using triangular lattice graphs, and the second using random graphs. In both cases, all graphs in the dataset are the same size (we use 10 and 20 node graphs). For the triangular lattice graph dataset, all graphs have the

Table 3: Relative dataset RMSE (%) for node localization, mean and standard deviation across 10 folds and 3 random parameter seeds.

| MODEL | EXPERIMENT | | | |
|---|---|---|---|---|
| | LATTICE (10) | LATTICE (20) | RANDOM (10) | RANDOM (20) |
| GCN (sync) | $20.8 \pm 1.0$ | $27.4 \pm 0.3$ | $27.1 \pm 0.5$ | $30.2 \pm 0.5$ |
| GCN (async) | $287.3 \pm 74.7$ | $964.2 \pm 227.7$ | $360.5 \pm 113.6$ | $143.8 \pm 30.3$ |
| IGNN | $27.9 \pm 0.6$ | $28.3 \pm 0.5$ | $30.2 \pm 0.5$ | $33.7 \pm 0.6$ |
| Energy GNN | $\mathbf{20.0 \pm 1.0}$ | $\mathbf{24.2 \pm 0.8}$ | $\mathbf{22.7 \pm 2.1}$ | $\mathbf{27.6 \pm 3.1}$ |

same structure but different permutations of node features, and node positions lie in the unit square. For the random graphs dataset, we sample points uniformly in the unit square and connect nodes by an edge if they are within a distance of 0.5. Each dataset has 500 graphs.

Table 3 shows relative test RMSE for each dataset. GCNs and IGNNs, neither of which use edge features, perform reasonably well for lattice graphs, where edge features are uninformative since distances to neighbors are constant. For random graphs, the edge features are necessary for localization, so GCNs and IGNNs perform more poorly. Energy GNNs achieve the best performance for all datasets, but by a small margin, as localization is a difficult task.

## 7 CONCLUSION AND FUTURE WORK

We believe GNNs have the potential to provide learning frameworks for distributed systems, with applications to privacy, robotics, remote sensing, and other domains. However, as we have articulated, most conventional GNN architectures are not compatible with asynchronous inference and this hinders their deployment on these types of problems. We reiterate that this is a distinct problem from distributed training, which has different constraints but still assumes synchronism at inference time. In this work, we identified some extant architectures which are robust to asynchrony, and presented a competitive, novel class in the form of energy GNNs. The guarantees of our method arise from framing inference as a convex optimization problem that is amenable to "hogwild" asynchronous techniques as in Recht et al. (2011). We evaluate the performance of energy GNNs on a number of synthetic tasks, motivated by the application of GNNs to decentralized, low-constrained multi-agent systems, where distributed, asynchronous inference is desirable. In these taks, we achieve better performance than IGNN Gu et al. (2020), another 'hogwild-able' GNN architecture. In addition to its robustness to asynchronism, our method is comparable in generalization performance (on benchmark datasets) with other modern GNN architectures that do not offer these guarantees.

We hope the positive results of our synthetic experiments motivates additional work in applying 'hogwild-able' GNN architectures to multi-agent related tasks. Inference over large graphs is another glaring application for asynchronous GNNs which should be explored in future work. This will likely also require distributed training, which will have to be adapted to the particulars of the forward and backward pass of asynchronous GNNs. We additionally believe it is crucial to explore asynchronism in training; this will involve additional distributed computation to solve the adjoint problem and collectively compute derivatives in a decentralized way. Another line of work which we expect to be interesting is real-time inference of *dynamic* graphs, both because of the relevance to problems in, e.g., robotics, but also due to the "anytime" nature of the energy GNN architecture. We are optimistic that the energy GNN framework itself, and in general the optimization-based view of GNN message passing, will provide a path forward for "learning to learn" distributed algorithms.

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
