# A APPENDIX

## A.1 PARTIALLY ASYNCHRONOUS ITERATIVE ALGORITHMS

Concretely, consider a collection of $n$ nodes, which could be, for example, a set of processors carrying out a distributed computation. Processor $i$ is allocated a block $\boldsymbol{x}_i \in \mathbb{R}^{d_i}$ of the length $d = \sum_{i=1}^{n} d_i$ block vector of problem data $\boldsymbol{x} = (\boldsymbol{x}_1, \boldsymbol{x}_2, \ldots, \boldsymbol{x}_n) \in \mathbb{R}^d$, and we aim to coordinate the nodes to execute an iterative algorithm using node-specific update functions $f_i : \mathbb{R}^d \mapsto \mathbb{R}^{d_i}$. Suppose we are given initial conditions $\boldsymbol{x}(t) \in \mathbb{R}^d$ for each $t \leq 0$, a set $T^i \subseteq \{0, 1, 2, \ldots\}$ of times at which the node $i$ is updated, and $\tau_j^i(t)$, $i, j = 1, \ldots, n$ for each $t \in T^i$. The quantities $s_{ij}(t) = t - \tau_j^i(t) \in [0, t]$ can be interpreted as the amount (in time) by which information associated with node $j$ is outdated or "stale" when used in the update of $\boldsymbol{x}_i$ at time $t$. This staleness value is bounded by $0$ since at best, $i$ executes an update with perfectly up to date information about node $j$ (corresponding to strict equality $\tau_j^i(t) = t$). At worst, node $i$ has received no data from node $j$, meaning node $i$ executes an update using the initial condition associated with node $j$ (corresponding to strict equality $\tau_j^i(t) = 0$).

The update equations describing the algorithm are, for $t \geq 0$:

$$\boldsymbol{x}_i(t+1) = \boldsymbol{x}_i(t), \quad \text{if } t \notin T^i, \tag{16}$$

$$\boldsymbol{x}_i(t+1) = f_i(\boldsymbol{x}_1(\tau_1^i(t)), \boldsymbol{x}_2(\tau_2^i(t)), \ldots, \boldsymbol{x}_n(\tau_n^i(t))), \quad \text{if } t \in T^i. \tag{17}$$

Partial asynchronism corresponds to the following assumptions:

1. For every node $i$ and for every $t \geq 0$, at least one of the elements of the set $\{t, t+1, \ldots, t+B-1\}$ belongs to $T^i$.

2. There holds
$$t - B < \tau_j^i(t) \leq t,$$
for all $i$ and $j$, and all $t \geq 0$ belonging to $T^i$.

3. There holds $\tau_i^i(t) = t$ for all $i$ and $t \in T^i$

In words, assumption 1 states that nodes are updated at least once every $B$ units of time, assumption 2 states that the staleness of information associated with any node is bounded by $B$ time units (conversely, information is periodically "purged" from the system after at most $B$ time units), and assumption 3 states that node $i$ maintains the latest version of $\boldsymbol{x}_i$. We assume this model of partial asynchrony in our work.

## A.2 INPUT-CONVEX GNN ARCHITECTURE DETAILS

As in Amos et al. (2017), we construct a parametric family of neural networks $f_\theta(x, y)$ with inputs $x \in \mathbf{R}^n, y \in \mathbf{R}^m$ which are convex with respect to $y$ (i.e. a subset of the inputs). We define a $k$-layer partially convex neural network by the recurrences:

$$u_{i+1} = \tilde{g}_i(\tilde{W}_i u_i + \tilde{b}_i)$$
$$z_{i+1} = g_i\big($$
$$W_i^{(z)}(z_i \circ [W_i^{(zu)}u_i + b_i^{(z)}]_+)+$$
$$W_i^{(y)}(y \circ (W_i^{(yu)}u_i + b_i^{(y)})+$$
$$W_i^{(u)}u_i + b_i\big)$$

$$f_\theta(x, y) = z_k, \quad u_0 = x, \quad z_0 = 0$$

Provided the $W^{(z)}$ are elementwise nonnegative, and the activation functions $g_i$ are non-decreasing in each argument, it follows that $f_\theta$ is convex in $y$.

In the context of an energy GNN, an energy layer is comprised of two ICNNs, specialized to operate on graph structures. The message function $m$ in eq. (12) corresponds with an ICNN which has node embeddings $\boldsymbol{h}_i, i = 1, \ldots, n$ and node features $\boldsymbol{X}$ as its convex and non-convex inputs, respectively. A second ICNN corresponds with the update function $u$ in eq. (12), which is convex in the node embeddings and the messages computed by the message function, and nonconvex in the node features. The graph energy can then be written as the sum of the outputs of the second updating ICNN.

## A.3 IMPLICIT DIFFERENTIATION

Since we use an optimization procedure to compute the node embeddings within the forward pass , we need to obtain derivatives of the node embeddings with respect to the parameters of the model. We compute derivatives by implicitly differentiating the optimality conditions, using the fact that at the solution of the energy minimization problem, we have parameters $\theta^* \in \mathbb{R}^p$ and node embeddings $\boldsymbol{H}^* \in \mathbb{R}^{n \times k}$ such that:

$$\frac{\partial E_{\theta^*}}{\partial \boldsymbol{H}}(\boldsymbol{H}^*) = \boldsymbol{0}. \tag{18}$$

Let $g(\boldsymbol{H}, \theta) = \frac{\partial E_\theta}{\partial \boldsymbol{H}}(\boldsymbol{H})$ and $h^*(\theta^*) = \boldsymbol{H}^*$, so that we can write the optimality conditions in terms of the parameters only:

$$g(h^*(\theta^*), \theta^*) = 0. \tag{19}$$

Given an objective $\mathcal{L} : \mathbb{R}^p \mapsto \mathbb{R}$, the desired quantity is the total derivative of $\mathcal{L}$ with respect to the parameters. By the chain rule,

$$\frac{d\mathcal{L}}{d\theta} = \frac{\partial \mathcal{L}}{\partial h^*} \frac{dh^*}{d\theta} + \frac{\partial \mathcal{L}}{\partial \theta}. \tag{20}$$

We compute $\frac{\partial \mathcal{L}}{\partial h^*}$ and $\frac{\partial \mathcal{L}}{\partial \theta}$ using normal automatic differentiation, and the solution Jacobian $\frac{dh^*}{d\theta}$ using implicit differentiation. Notice that, at the fixed point (where the optimality constraint is satisfied), we have:

$$\frac{d}{d\theta} g(h^*(\theta), \theta) = \boldsymbol{0} \tag{21}$$

$$\frac{\partial g}{\partial h^*} \frac{dh^*}{d\theta} + \frac{\partial g}{\partial \theta} = \boldsymbol{0} \tag{22}$$

$$\frac{\partial g}{\partial h^*} \frac{dh^*}{d\theta} = -\frac{\partial g}{\partial \theta}. \tag{23}$$

This is the primal (or tangent) system associated with the constraint function $g$. In our setup we utilize reverse mode automatic differentiation, since $p \gg 1$ parameters are mapped to a single scalar objective. Provided $\frac{\partial g}{\partial h^*}$ is invertible, we can rewrite the solution Jacobian as:

$$\frac{dh^*}{d\theta} = -\left(\frac{\partial g}{\partial h^*}\right)^{-1} \frac{\partial g}{\partial \theta}, \tag{24}$$

and substitute this expression into eq. (20) as follows:

$$\frac{d\mathcal{L}}{d\theta} = -\frac{\partial \mathcal{L}}{\partial h^*}\left(\frac{\partial g}{\partial h^*}\right)^{-1} \frac{\partial g}{\partial \theta} + \frac{\partial \mathcal{L}}{\partial \theta}. \tag{25}$$

For reverse mode, we compute the dual (or adjoint) of this equation,

$$\frac{d\mathcal{L}}{d\theta}^T = -\frac{\partial g}{\partial \theta}^T \left(\frac{\partial g}{\partial h^*}\right)^{-T} \frac{\partial \mathcal{L}}{\partial h^*}^T + \frac{\partial \mathcal{L}}{\partial \theta}^T, \tag{26}$$

And solve the dual system:

$$\frac{\partial g}{\partial h^*}^T \lambda = -\frac{\partial \mathcal{L}}{\partial h^*}^T \tag{27}$$

for the dual variable $\lambda$.

## A.4 Provided the step size is small enough, the asynchronous energy GNN inference algorithm converges

To prove that the energy GNN inference algorithm converges under the assumption of partial asynchronism, it is useful to frame the method as an unconstrained gradient method. Provided the step size is small enough (which will be made precise), gradient algorithms converge under partial asynchronism (Bertsekas, 1983). Gradient algorithms operating in asynchronous frameworks have been analyzed in the asynchronous algorithms literature dating back to the late 1960s. We cite a proof given by (Bertsekas & Tsitsiklis, 1989), and simply show that the assumptions required by **Proposition 5.1** holds for our algorithm, and thus it converges.

First, we build on appendix A.1 and section 5.3 to establish that the asynchronous EGNN inference algorithm corresponds to solving an optimization problem via a gradient method. In eq. (12) we introduced a scalar-valued convex "energy" function $e_i : \mathbb{R}^d \mapsto \mathbb{R}$ associated with each node $i = 1, \dots, n$. The dimension $d$ of **dom** $e_i$ arose from our setup in section 5.3, i.e., concatenating the graph data into a block-vector for correspondence with the partial asynchronism model. But recall that $e_i$ does not depend on the data of any node which is not a neighbor of node $i$, which is to say,

$$j \notin \text{ne}(i) \implies \frac{\partial e_i}{\partial \boldsymbol{x}_j}(y) = 0 \text{ for all } y \in \mathbb{R}^{d_j}.$$

We also noted in that section that, in fact, the energy associated with node $i$ depends only on the latent values associated with node $i$ and its neighbors: $\{\boldsymbol{h}_i\} \cup \{\boldsymbol{h}_j \mid (i, j) \in \mathcal{E}\}$. Here we overload $e_i$ and $\boldsymbol{x}_i$ so that the domain of $e_i$ does not include these irrelevant auxiliary variables. In what follows, we have **dom** $e_i = \mathbb{R}^{d_i}$, with $d_i = k(|\text{ne}(i)| + 1)$: the length of the latent $\boldsymbol{h}_i \in \mathbb{R}^k$ associated with node $i$, plus the length of the latents $\text{ne}(\boldsymbol{h}_i) = \{\boldsymbol{h}_j | (i, j) \in \mathcal{E}\}$ associated with the neighbors of node $i$. Let $\boldsymbol{x}_i = \{\boldsymbol{h}_i\} \cup \{\boldsymbol{h}_j \mid (i, j) \in \mathcal{E}\} \in \mathbb{R}^{d_i}$. In words, the energy of node $i$ depends on $\boldsymbol{x}_i$ which is comprised of its own latent value, and the latent values of its neighbors. Generalizing the notion of an "energy function" to the entire graph, we denote the "graph energy" $E : \mathbb{R}^{nk} \mapsto \mathbb{R}$ as the sum of the node energy functions:

$$E(\boldsymbol{h}_1, \boldsymbol{h}_2, \dots, \boldsymbol{h}_n) = \sum_{i=1}^{n} e_i(\boldsymbol{x}_i).$$

Our asynchronous inference algorithm operates on a block vector $\boldsymbol{h}(t) \in \mathbb{R}^{nk}$ whose $i$th block (the latent associated with node $i$), is updated by node $i$ according to

$$\boldsymbol{h}_i(t + 1) = \boldsymbol{h}_i(t) + \alpha \boldsymbol{s}_i(t), \quad i = 1, \dots, n,$$

with $\alpha > 0$ a step size and $\boldsymbol{s}_i(t) \in \mathbb{R}^k$ the update direction. Per our overview of partial asynchronism in appendix A.1, let $T^i$ be the set of times when node $i$ performs an update. We assume that

$$\boldsymbol{s}_i(t) = 0, \quad \forall t \notin T^i.$$

Node $i$ retains possibly outdated information about the latents associated with the other nodes in the graph: importantly, it may have stale information about its neighbors. Denote $\boldsymbol{h}^i(t) \in \mathbb{R}^{nk}$ as node $i$'s view of the graph at time $t$. That is,

$$\boldsymbol{h}^i(t) = (\boldsymbol{h}_1(\tau_1^i(t)), \dots, \boldsymbol{h}_n(\tau_n^i(t))).$$

We make the same assumptions about the $\tau_j^i(t)$ as above. With this setup and notation in hand, we simply confirm that the assumptions associated with Proposition 5.1 hold, which proves convergence.

### A.4.1 Assumption 5.1

Since both convexity and absolute continuity are preserved under nonnegative summation, the graph energy $E$ is a smooth, strictly convex function. Without loss of generality, we can assume $e_i(\boldsymbol{x}_i) \geq 0$ for all $\boldsymbol{x}_i \in \mathbb{R}^{d_i}$. This follows because the $e_i$ are strictly convex, therefore the optimal value $p_i^* = \inf\{e_i(\boldsymbol{x}_i)\}$ is achieved and thus $\tilde{e}_i = e_i + p_i^*$ is nonnegative. Thus, we have that the graph energy is the sum of nonnegative terms: $E(\boldsymbol{h}_1, \boldsymbol{h}_2, \dots, \boldsymbol{h}_n) \geq 0$ for all $(\boldsymbol{h}_1, \boldsymbol{h}_2, \dots, \boldsymbol{h}_n) \in \mathbb{R}^{nk}$.

### A.4.2 Assumption 5.2

For those times $t \in T^i$, our update direction for node $i$ is $\boldsymbol{s}_i(t) = -\nabla_{\boldsymbol{h}_i} E(\boldsymbol{h}^i(t))$, thus $\boldsymbol{s}_i(t)^T \nabla_{\boldsymbol{h}_i} E(\boldsymbol{h}^i(t)) \leq 0$, and our algorithm satisfies part (a). In words, the update direction is

such that the graph energy does not increase. Further, we have that $|\boldsymbol{s}_i(t)| = |\nabla_i E(\boldsymbol{h}^i(t))|$ which means part (b) holds with $K_2 = K_3 = 1$.

Since assumption 5.1 and 5.2 hold, and our inference algorithm is partially asynchronous, proposition 5.1 holds, and convergence is guaranteed provided the stepsize $\alpha$ is small enough. In particular, the maximum stepsize is:

$$\alpha_0 = \frac{1}{1 + (n+1)B},$$

where $B > 0$ is the positive integer bounding staleness as above. This is intuitive: as noted in (Bertsekas & Tsitsiklis, 1989), if the nodes make larger steps, they must inform their neighbors more often. We simply summarize the conclusion of proposition 5.1 here in the context of our algorithm: we have that $\lim_{t\to\infty} \nabla E(\boldsymbol{h}(t)) = 0$ provided $0 < \alpha < \alpha_0$, with $\alpha_0$ defined above.

## A.5 ASYNCHRONOUS GNN IMPLEMENTATION

In our experiments with asynchronous inference, we simulate partial asynchrony [1]. We set the maximum staleness bound $B = 20$. Our implementation ensures nodes never deviate from one another in terms of number of updates by more than 1, i.e., all nodes update at a regular frequency.

---

**Algorithm 1** Simulated asyncronous GNN inference

---

Initialize each node in $\mathcal{G}$ with all GNN parameters, its own node features $\boldsymbol{X}_i$, and for each of its neighbors $j$, the node and edge features $\boldsymbol{X}_j$, $\boldsymbol{E}_{ij}$, and weights $\boldsymbol{A}_{i,j}$. Initialize the current iteration count $t_i = 0$ for all nodes $i$. Let $\tau_j^i$ be the number of iterations by which node $i$'s view of node $j$ is outdated; we maintain a global view of these values for all neighboring nodes pairs and control the staleness value in simulation. Let $L$ be the number of layers in the GNN, equal to $\infty$ for infinite-depth GNNs. Let $T$ be the total number of simulated node updates. Let $n$ be the number of nodes in the graph. Let $B$ be the maximum staleness of a node's view of its neighbor values.

ordering = [ ]
**for** $t = 0, \ldots, T$ **do**
    **if** $t \mod n = 0$ **then**
        Set ordering as a random permutation of the node indices
    **end if**
    $i = \text{ordering}[t \mod n]$
    **if** $t_i < L$ **then**
        **for** neighbors $j$ of node $i$ **do**
            Sample an updated staleness for node $j$, $(\tau_j^i)' \sim \text{Uniform}(0, \max(\tau_j^i, B))$
            $\tau_j^i \leftarrow (\tau_j^i)'$

        **end for**
        $\boldsymbol{m}_i = g^{t_i}\left(\{m^{t_i}(\boldsymbol{h}_j(\tau_j^i), \boldsymbol{h}_i, \boldsymbol{X}_j, \boldsymbol{X}_i, \boldsymbol{E}_{ij}; \theta_m^{t_i}) \mid j \in \text{ne}(i)\}, \boldsymbol{A}_i; \theta_g^{t_i}\right)$
        $\boldsymbol{h}_i \leftarrow u^{t_i}\left(\boldsymbol{m}_i, \boldsymbol{h}_i, \boldsymbol{X}_i; \theta_u^{t_i}\right)$
        $t_i \leftarrow t_i + 1$
        $\tau_i^j \leftarrow \tau_i^j + 1 \qquad \forall j \in \text{ne}(i)$
    **else**
        $\hat{y}_i = o_\phi(h_i^L)$
    **end if**
**end for**

---

## A.6 EXPERIMENT DETAILS (SYNTHETIC EXPERIMENTS)

For all experiments, we use 2 layers of message passing for GCN, and for IGNN and energy GNN we use a single parameterized layer of message passing.

For training all models we use the Adam optimizer with weight decay. We record the test performance at the epoch corresponding to the best training loss. For asynchronous inference, we set the maximum staleness of information $B = 20$, and follow the asynchronous inference simulation presented in Appendix A.5.

For energy GNNs, we define the architecture of the energy function $E_\theta$, which is a PICGNN, as some variant of the following:

$$\boldsymbol{m}_{ij} := m_n(\boldsymbol{h}_j, \boldsymbol{h}_i, \boldsymbol{X}_j, \boldsymbol{X}_i \boldsymbol{E}_{ij}; \theta_{mn}) \quad \forall i, j \in E \tag{28}$$

$$\boldsymbol{m}_{ii} := m_s(\boldsymbol{h}_i, \boldsymbol{X}_i; \theta_{ms}) \quad \forall i \in V \tag{29}$$

$$\boldsymbol{m}_i := \zeta \left( \sum_{j \in \text{ne}(i) \cup i} \boldsymbol{m}_{ij} \right) \tag{30}$$

$$_i := u\left(\boldsymbol{m}_i, \boldsymbol{h}_i, \boldsymbol{X}_i; \theta_u\right) + \beta ||\boldsymbol{h}_i||_2^2 \tag{31}$$

$$\tag{32}$$

where $m_n$ is a message function applied to information pertaining to neighbors (with parameters $\theta_{mn}$) and $m_s$ is a message function applied to a node's own information (with parameters $\theta_{ms}$). In other words, we introduce an asymmetric self loop where each node is in it's own neighborhood and a separate message function is used to transform its data before aggregation. We use a 3-layer PICNN for $m_n$, $m_s$, and $u$, where convex and non-convex inputs are grouped and concatenated prior to input. We use L-BFGS to minimize the energy during training.

Consistent with Gu et al. (2020), for IGNNs, we always do aggregation with the symmetric renormalized adjacency matrix with added self loops. The IGNN has a single parameterized layer of message passing. All GCNs in our experiments use two layers.

We set the node embedding sizes such that the parameter counts are approximately equal for all models.

### A.6.1 CHAINS

For all architectures, the node output net $o_\phi$ is an affine function. We optimize the loss (binary cross entropy) with learning rate $0.001$, and a decay rate of $0.9$ every $150$ epochs. We train for a maximum of $2000$ epochs.

For energy GNNs, we set $\boldsymbol{h}_i \in \mathbf{R}^2$. We use a 3-layer PICNN for $m_n$, $m_s$, and $u$ with layer sizes $(4, 4, 2)$, $(4, 4, 2)$, and $(4, 4, 1)$, where the inputs to each networks are $(\boldsymbol{h}_j, \boldsymbol{h}_i, \boldsymbol{X}_i, \boldsymbol{X}_j), (\boldsymbol{h}_i, \boldsymbol{X}_i), (\boldsymbol{m}_i, \boldsymbol{h}_i)$, respectively. We set $\beta = 0.1$. For IGNN, we set $\boldsymbol{h}_i \in \mathbf{R}^{22}$, and for GCN, we set $\boldsymbol{h}_i \in \mathbf{R}^{16}$.

### A.6.2 COUNTING/SUMMING

For all architectures, the node output net $o_\phi$ is a 3 layer neural network with layer sizes $(4, 4, 1)$. We optimize the loss (mean squared error) with learning rate $0.0025$, and a decay rate of $0.9$ every $200$ epochs. We train for a maximum of $5000$ epochs.

For each dataset, we scale the prediction target based on the largest graph size, so the prediction target falls in the range $[0, 1]$.

For energy GNNs, we set $\boldsymbol{h}_i \in \mathbf{R}^1$. We use a 3-layer PICNN for $m_n$, $m_s$, and $u$ with layer sizes $(4, 4, 2)$, $(4, 4, 2)$, and $(4, 4, 1)$. For counting, the inputs to each of the networks are $(\boldsymbol{h}_j, \boldsymbol{h}_i), (\boldsymbol{h}_i), (\boldsymbol{m}_i, \boldsymbol{h}_i)$, respectively. For summing, the inputs to each networks are $(\boldsymbol{h}_j, \boldsymbol{h}_i, \boldsymbol{X}_i, \boldsymbol{X}_j), (\boldsymbol{h}_i, \boldsymbol{X}_i), (\boldsymbol{m}_i, \boldsymbol{h}_i)$, respectively. We set $\beta = 0.02$. For counting, we set $\boldsymbol{h}_i \in \mathbf{R}^{10}$ for IGNN and $\boldsymbol{h}_i \in \mathbf{R}^8$ for GCN. For counting, we set $\boldsymbol{h}_i \in \mathbf{R}^{18}$ for IGNN and $\boldsymbol{h}_i \in \mathbf{R}^{13}$ for GCN.

### A.6.3 COORDINATES

For all architectures, the node output net $o_\phi$ is a 2 layer neural network with layer output sizes $(2, 2)$. We optimize the loss (mean squared error) with learning rate $0.001$, and a decay rate of $0.99$ every $150$ epochs. We train for a maximum of $5000$ epochs.

For energy GNNs, we set $\boldsymbol{h}_i \in \mathbf{R}^2$. We use a 3-layer PICNN for $m_n$, $m_s$, and $u$ with layer sizes $(6, 6, 4)$, $(6, 6, 2)$, and $(4, 4, 1)$. The inputs to each of the networks are $(\boldsymbol{h}_j, \boldsymbol{h}_i, \boldsymbol{X}_i, \boldsymbol{X}_j, \boldsymbol{E}_{ij}), (\boldsymbol{h}_i, \boldsymbol{X}_i), (\boldsymbol{m}_i, \boldsymbol{h}_i)$, respectively. We set $\beta = 0.1$. For the lattice and random graph datasets consisting of graphs with 10 nodes (and thus using node features $\boldsymbol{X} \in \mathbb{R}^{n \times 10}$), we set $\boldsymbol{h}_i \in \mathbf{R}^{26}$ for IGNN and $\boldsymbol{h}_i \in \mathbf{R}^{19}$ for GCN. For the datasets with graphs of size 20, we set $\boldsymbol{h}_i \in \mathbf{R}^{22}$ for IGNN and $\boldsymbol{h}_i \in \mathbf{R}^{18}$ for GCN.

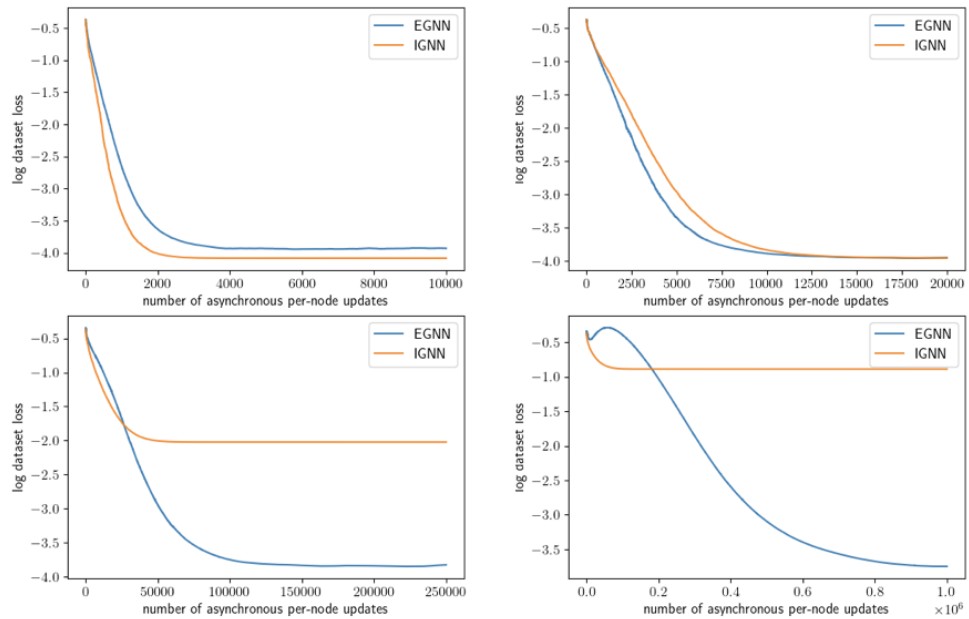

Figure 2: Log dataset loss during asyncronous inference for chains experiment. From left to right and top to bottom, plots correspond to chains of length 10, 20, 50, 100.

## A.7 PARTIALLY ASYNCRONOUS IGNN/ENERGY GNN INFERENCE PLOTS

In Figure 2, we show plots of dataset loss over time during partially asynchronous inference of energy GNNs (referred to as EGNN) and IGNNs for the synthetic chains experiments. The loss achieved by asynchronous inference converges given enough iterations for both architectures, equivalently showing that node embeddings eventually converge.

### A.8 BENCHMARK DATASETS

For all experiments with benchmark datasets, we use 2 layers of message passing for GCN, and for IGNN and energy GNN we use a single parameterized layer of message passing.

#### A.8.1 DATASET DETAILS

The benchmark datasets we report performance for are MUTAG and PROTEINS, where the prediction task is graph classification, and PPI, where the prediction task is node classification.

**MUTAG** MUTAG is a dataset consisting of 188 graphs, each of which corresponds to a nitroaromatic compound (Srinivasan et al., 1996). The goal is to predict the mutagenicity of each compound on *Salmonella typhimurium*. Nodes in the graphs correspond to atoms (and are associated with a one-hot encoded feature in $\mathbb{R}^7$ corresponding to the atom type), and edges correspond to bonds. The average number of nodes in a graph is 17.93, and the average number of edges is 19.79.

**PROTEINS** The PROTEINS dataset Borgwardt et al. (2005) consists of 1113 graphs, each of which corresponds to a protein. The task is predicting whether or not the protein is an enzyme. Nodes in the graph correspond to amino acids in the protein (and are associated with node features in $\mathbb{R}^3$ representing amino acid properties). Edges connect amino acids that are less than some threshold distance from one another in the protein. The average number of nodes in a graph is 39.06, and the average number of edges is 72.82.

**PPI** The PPI dataset (Hamilton, 2020) consists of 24 graphs, each of which corresponds to a protein-protein interaction network found in different areas of the body. Each node in the graph corresponds to a protein, with edges connecting proteins that interact with one another. Nodes are associated with features in $\mathbb{R}^{50}$, representing some properties of the protein. Each protein has 121 binary prediction targets, each of which corresponds to some ontological property that the protein may or may not have.

#### A.8.2 GRAPH CLASSIFICATION

We test energy GNNs on two binary graph classification benchmark datasets; MUTAG and PROTEINS.

For MUTAG and PROTEINS, the node output net $o_\phi$ is a 2 layer neural network with layer output sizes $(2, 2)$ for all architectures. We take the mean of the node predictions, and use these as input to a 2 layer neural network with layer output sizes $(2, 1)$ to obtain the graph level prediction. We optimize the loss (binary cross entropy) with learning rate $0.001$, and a decay rate of $0.95$ every $250$ epochs. We train for a maximum of $5000$ epochs for MUTAG and $2500$ epochs for PROTEINS.

For energy GNNs, we set $\boldsymbol{h}_i \in \mathbf{R}^2$. We use a 3-layer PICNN for $m_n$, $m_s$, and $u$ with layer sizes $(4, 4, 2)$, $(4, 4, 2)$, and $(4, 4, 1)$. The inputs to each of the networks are $(\boldsymbol{h}_j, \boldsymbol{h}_i, \boldsymbol{X}_i, \boldsymbol{X}_j), (\boldsymbol{h}_i, \boldsymbol{X}_i), (\boldsymbol{m}_i, \boldsymbol{h}_i)$, respectively. We set $\beta = 0.05$. For MUTAG we set $\boldsymbol{h}_i \in \mathbf{R}^7$ for IGNN and $\boldsymbol{h}_i \in \mathbf{R}^{15}$ for GCN. For PROTEINS we set $\boldsymbol{h}_i \in \mathbf{R}^{10}$ for IGNN and $\boldsymbol{h}_i \in \mathbf{R}^{14}$ for GCN.

We perform 10-fold cross validation and report average classification accuracy and standard deviations in Table 4. We include performance reported by Gu et al. (2020) (marked by an asterisk), as well as performance of our own implementation of IGNN with a single layer and GCN with two layers.

#### A.8.3 NODE CLASSIFICATION

For node classification, we consider the PPI dataset. We use a 20/2/2 train/valid/test split consistent with Hamilton et al. (2017).

For all architectures, the node output net $o_\phi$ is a 2 layer neural network with layer output sizes $(8, 121)$. We optimize the loss (binary cross entropy) with learning rate $0.005$, and a decay rate of $0.9$ every $200$ epochs. We train for a maximum of $2000$ epochs.

For energy GNNs, we set $\boldsymbol{h}_i \in \mathbf{R}^2$. We use a 3-layer PICNN for $m_n$, $m_s$, and $u$ with layer sizes $(4, 4, 2)$, $(4, 4, 2)$, and $(4, 4, 1)$. The inputs to each of the networks are

Table 4: Graph classification accuracy (%). Results are averaged (and standard deviations are computed) using 10 fold cross validation. Asterisked values are from Gu et al. (2020).

|  | DATASET | |
| --- | --- | --- |
| **MODEL** | **MUTAG** | **PROTEINS** |
| GCN* (5 layer) | $85.6 \pm 5.8$ | $76.0 \pm 3.2$ |
| IGNN* (3 layer) | $89.3 \pm 6.7$ | $77.7 \pm 3.4$ |
| GCN (2 layer) | $72.83 \pm 6.6$ | $70.89 \pm 3.2$ |
| IGNN (1 layer) | $75.47 \pm 7.3$ | $70.98 \pm 2.8$ |
| Energy GNN | $87.78 \pm 5.3$ | $71.25 \pm 2.6$ |

$(\boldsymbol{h}_j, \boldsymbol{h}_i, \boldsymbol{X}_i, \boldsymbol{X}_j), (\boldsymbol{h}_i, \boldsymbol{X}_i), (\boldsymbol{m}_i, \boldsymbol{h}_i)$, respectively. We set $\beta = 0.05$. We set $\boldsymbol{h}_i \in \mathbf{R}^{10}$ for IGNN and $\boldsymbol{h}_i \in \mathbf{R}^{10}$ for GCN.

Table 5 shows average micro-f1 scores for energy GNNs compared to other GNN architectures. We include performance reported by Gu et al. (2020) (marked by an asterisk), as well as performance of our own implementation of IGNN with a single layer and GCN with two layers. Where layer specifications are not included (for asterisked values), we were unable to determine them from Gu et al. (2020).

Table 5: Node classification accuracy on PPI dataset (%). Results are averaged (and std are computed) using 10 fold cross validation. Asterisked values are from Gu et al. (2020).

| METHOD | micro f1 |
| --- | --- |
| MLP* | 46.2 |
| GCN* | 59.2 |
| GraphSAGE* | 78.6 |
| GAT* (3 layer) | 97.3 |
| IGNN* (5 layer) | 97.6 |
| IGNN (1 layer) | 76.8 |
| GCN (2 layer) | 76.7 |
| Energy GNN | 76.2 |

## A.9 EIGNN UPDATE AS OPTIMIZATION

Previous work (Yang et al., 2021; Zhu et al., 2021) has shown that many GNN node embedding update functions correspond to optimization of some objective function. This is highly related to energy GNNs; however, the class of objective functions considered is limited to a specific form. We show the optimzation view of EIGNN Liu et al. (2021) as an example, and refer the reader to (Yang et al., 2021; Zhu et al., 2021) for a more comprehensive overview. We note that the optimization objective functions for all GNNs studied in (Yang et al., 2021; Zhu et al., 2021) have a form similar to that for EIGNN, which as we show, consists of (1) a quadratic penalty between some function of the input features and the node embeddings and (2) a possibly non-convex, graph-aware penalty term that encourages smoothness of node embeddings across edges.

The EIGNN update is defined as follows:

$$\boldsymbol{H}^{t+1} = \gamma\alpha\tilde{\boldsymbol{A}}\boldsymbol{H}^t\boldsymbol{F}^T\boldsymbol{F} + \boldsymbol{X}. \tag{33}$$

The matrix $\tilde{\boldsymbol{A}} = (\boldsymbol{D}+\boldsymbol{I})^{-1/2}(\boldsymbol{A}+\boldsymbol{I})(\boldsymbol{D}+\boldsymbol{I})^{-1/2}$ is called the *symmetric renormalized adjacency matrix*. The matrix $\boldsymbol{F} \in \mathbb{R}^{k\times k}$ is comprised of parameters, $\alpha > 0$ is a scaling factor equal to $\frac{1}{||\boldsymbol{F}^T\boldsymbol{F}||_F+\epsilon}$ with arbitrarily small $\epsilon$, and $\gamma \in (0,1]$ is an additional scaling factor. The overall scaling factor $\gamma\alpha$ is chosen to ensure that the update is contractive, from which it follows that the sequence of iterates converges.

**Proposition 1**. The fixed point $\boldsymbol{H}^*$ achieved by the EIGNN update function satisfying $\boldsymbol{H}^* = \alpha\tilde{\boldsymbol{A}}\boldsymbol{H}^*\boldsymbol{F}^T\boldsymbol{F} + \boldsymbol{X}$ corresponds to the minimum of the following convex energy function:

$$E(\boldsymbol{F}, \boldsymbol{H}) = \frac{1}{2}||\boldsymbol{H}(\boldsymbol{I}-\alpha\boldsymbol{F}^T\boldsymbol{F})^{-1/2} - \boldsymbol{X}(\boldsymbol{I}-\alpha\boldsymbol{F}^T\boldsymbol{F})^{-1}(\boldsymbol{I}-\alpha\boldsymbol{F}^T\boldsymbol{F})^{-1/2}||_F^2 + \tag{34}$$

$$\frac{\alpha}{2}\text{trace}(\boldsymbol{H}^T(\boldsymbol{I}-\gamma\tilde{\boldsymbol{A}})\boldsymbol{H}\boldsymbol{F}^T\boldsymbol{F})$$

$$= \frac{1}{2}||\boldsymbol{H}\boldsymbol{W} - \boldsymbol{X}(\boldsymbol{W}\boldsymbol{W})^{-1}\boldsymbol{W}||_F^2 + \frac{\alpha}{2}\text{trace}(\boldsymbol{H}^T(\boldsymbol{I}-\gamma\tilde{\boldsymbol{A}})\boldsymbol{H}\boldsymbol{F}^T\boldsymbol{F}) \tag{35}$$

where $\boldsymbol{W} = \boldsymbol{W}^T = (\boldsymbol{I}-\alpha\boldsymbol{F}^T\boldsymbol{F})^{-1/2}$ and $\gamma < 1/\lambda_{max}(\tilde{\boldsymbol{A}}) > 1/2$.

The Jacobian of the energy with respect to $\boldsymbol{H}$ is:

$$\frac{\partial E}{\partial \boldsymbol{H}} = (\boldsymbol{H}\boldsymbol{W} - \boldsymbol{X}(\boldsymbol{W}\boldsymbol{W})^{-1}\boldsymbol{W})\boldsymbol{W} + \alpha(\boldsymbol{I}-\gamma\tilde{\boldsymbol{A}})\boldsymbol{H}\boldsymbol{F}^T\boldsymbol{F}$$

$$= \boldsymbol{H}\boldsymbol{W}\boldsymbol{W} - \boldsymbol{X} + \alpha\boldsymbol{H}\boldsymbol{F}^T\boldsymbol{F} - \gamma\alpha\tilde{\boldsymbol{A}}\boldsymbol{H}\boldsymbol{F}^T\boldsymbol{F}$$

$$= \boldsymbol{H}(\boldsymbol{I}-\alpha\boldsymbol{F}^T\boldsymbol{F}) - \boldsymbol{X} + \alpha\boldsymbol{H}\boldsymbol{F}^T\boldsymbol{F} - \gamma\alpha\tilde{\boldsymbol{A}}^T\boldsymbol{F}$$

$$= \boldsymbol{H} - \boldsymbol{X} - \gamma\alpha\tilde{\boldsymbol{A}}\boldsymbol{H}\boldsymbol{F}^T\boldsymbol{F} \tag{36}$$

At the energy minimum, we have:

$$\boldsymbol{H}^* = \gamma\alpha\tilde{\boldsymbol{A}}\boldsymbol{H}^*\boldsymbol{F}^T\boldsymbol{F} + \boldsymbol{X}, \tag{37}$$

and thus recover the EIGNN fixed point. Note that Liu et al. (2021) appear to assume that the maximum eigenvalue of $\tilde{\boldsymbol{A}}$ satisfies $\lambda_{max}(\tilde{\boldsymbol{A}}) \leq 1$ and thus restrict $\gamma \in (0,1]$. However, this is false if $\tilde{\boldsymbol{A}}$ is equal to the symmetric renormalized adjacency matrix with added self loops, as the maximum eigenvalue may exceed 1. In this scenario, a choice of $\gamma > 1/\lambda_{max}(\tilde{\boldsymbol{A}})$ would result in a divergent sequence of iterates. We modify the restriction on gamma so that $\gamma \leq 1/\lambda_{max}(\tilde{\boldsymbol{A}})$ is satisfied.

## A.10 EIGNN Sensitivity to Inference via Closed Form Solution

EIGNNs Liu et al. (2021) are implicit GNNs employing contractive node embedding updates, where a closed form solution can be obtained instead of iterating the fixed point function. We find that performance of EIGNNs relies heavily on this closed form solution. When training EIGNNs with an iterative forward pass, performance is significantly diminished; we demonstrate this in Table 6 on the chains dataset. Furthermore, even if the model is trained using the closed form forward pass, at inference time, iterating on the fixed point function yields significantly different results; we demonstrate this in Table 6 on the chains dataset. In summary, even though EIGNNs are provably robust to distributed, asynchronous inference, their performance relies heavily on the closed form forward pass, which requires global information about the graph and cannot be easily performed in a distributed manner.

Table 6: EIGNN dataset accuracy on chains dataset under different training and inference regimes

| | Train/Test regime | | |
|---|---|---|---|
| chain length | closed form train / closed form test | closed form train / iterative test | iterative train / iterative test |
| 10 | 100 | 90 | 100 |
| 20 | 100 | 77.5 | 100 |
| 50 | 100 | 64 | 73 |
| 100 | 100 | 55.5 | 62.5 |