# OpenReview forum: "Graph Neural Networks Gone Hogwild"
_ICLR.cc/2024/Conference — Submitted to ICLR 2024_

### Official Review · Reviewer_fGVf · 2023-10-30

**Soundness:** 2 fair
**Presentation:** 2 fair
**Contribution:** 2 fair
**Rating:** 3
**Confidence:** 3

**Summary:**

The authors study the execution of graph neural network under asynchrony. This is claimed to be advantageous for executing GNNs on very large graphs or efficiently simulating a group of agents with limited communication. Specifically, the partial asynchronism model of execution is used, where the the authors propose a new architecture: Energy GNN. EGNN is an Implicit GNN with appropriate input-convex networks, for which the author can guarantee convergence of the EGNN under partial asynchronism (with appropriate step size). Moreover, the proposed method is empirically tested on a variety of synthetic and real world tasks.

**Strengths:**

The main idea of studying GNNs under a different perspective than the common synchronous message passing, in this case, the partial asynchrous execution model is both novel and interesting.

The theoretical analysis and the needed mathematical tools to properly connect the existing insights to graph neural networks are non-trivial and interesting contributions.

**Weaknesses:**

I do not see sufficient evidence to support the main claim of the paper for the mentioned use cases of more efficient execution on larger graphs. In its current form, there are no runtime measures, nor any comparisons where asynchronous EGNN is better (i.e. number of executed steps).

The experimental section should be improved. The choice of baselines against which EGNN is compared to can be improved. Furthermore, it is not exactly clear to me why it is these synthetic tasks the authors focus on - they are mainly long range communication tasks, which I find interesting, but are not in the spirit or in alignment with the main claims of the paper. Moreover, if the focus is more on the long range communication ability, I believe there should be other baselines (from the MPNNs) other than a 2 layer GCN.

Presentation of the paper could be improved. This not as big of a concern as the two points mentioned above and I believe to be (easily) fixable as they are not conceptual issues.

**Questions:**

Regarding the performance of synchronous GNNs which are executed asynchronously: I can see that the error can become arbitrarily large (as the synchronous assumption is violated). Are there other methods which could make them more robust? Could you already train them using the outlined schedule in Appendix A.5 (or dropout?) or use something similar to the original Scarselli work to encourage a contraction operation that converges? Especially a baseline similar to the Scarselli work I believe to be very important for your comparisons - this would support the claim that these models are too simple.

What about other MPNN baselines beyond the 2 layer GCN (which of course can learn to solve the tasks)? Do regular MPNNs which could learn the task correctly (and do so) also have the same performance drop?

If long range communication should be more of a main feature of the presented architecture I would recommend comparison against other baselines which also focus on this, i.e. by building deeper GNNs (or even graph transformers)

How long do the implicit GNNs and the EGNN run in practice? How deep are the “infinite depth” gnns in practice?

I was not quite able to follow the task setup for 6.3. I think a precise formulation of the problem using mathematical notation would help to clarify the task and training objective (can also be included in the Appendix).

If the main claim of the work is regarding improved performance on larger graphs when execution can be asynchronous I would expect at least some real world measurements or an otherwise adequate comparison (maybe number of node activations, or rounds per node) to further support the statement.

Related to the previous point, in Figure 2 of the Appendix: (i) what exactly is the dataset loss? And I interpret the plot that you need 2k updates per node in a graph of size 10 to reach convergence during inference? Do I understand this correctly and if this is the case, how is this more efficient than a 10 layer synchronous GNN?

In the real world dataset experiments EGNN performs worse than IGNN. Especially for PPI, EGNN only reaches performance similar to a 2 layer GCN, whereas EGNN is almost perfect. A performance loss, which comes at the benefit of execution speed would be ok or in some cases even desired. I would appreciate at least a comment or hypothesis on why the performance does not translate. Also I would suggest keeping the notation consistent and report the number of layers for all models in the tables.

The insight that EGNN still converges (with the right step size) seems like the main contribution of your work and I would suggest highlighting this more (in the main part).

The required step size is in O(1/nb) (which seems quite small). Could you provide an intuition on why this has to be dependent on n, to me this is not obvious (I would have thought maybe a graph parameter such as the node degree might be necessary?)

The results provided in the main part of the paper are only on individual modes and not reported as the mean over multiple runs if I understand correctly? I see that it might not be straightforward, as you also need to avg. over the used schedule - but I would appreciate it if the results would be averaged over multiple learned models.

What is the limiting factor of scaling to larger graphs? Could you go beyond the size of 100 nodes used in the paper, what are the tradeoffs?

Are the models trained on the same graph sizes that they are run on during inference? Or do you only train on one and then generalize across sizes for the synthetic tasks?

Additional Feedback:
I know that some of these comments are just personal preference, feel free to dismiss the comments that concern presentation style if you disagree.
In the related work section, I would encourage to include the work on asynchronous GNNs (rather than the focus on distr. training which is not that close to the presented ideas). This includes the work of Faber et al. (https://arxiv.org/abs/2205.12245) and Dudzik et al (https://arxiv.org/abs/2306.15632)
I would suggest to move the first two paragraphs before 3.1 and the GCN formulation in your notation to the Appendix. Instead, I would try to include A1, which actually outlines the specific assumptions of partial asynchronism.
The update of (9) should consider multisets instead of just regular sets right?
It is a bit weird to have all the def. on node levels but then (6) on graph level
Section title 4 is a bit long, also title A4 is almost a paragraph in itself.
I have trouble understanding the definition of $\tau^i_j$, shouldn’t it be the last updated timestep rather than the amount by which the embedding is outdated (this in my understanding is the quantity $s_ij$
In Section 4 adaptivity to dynamic inputs is mentioned (as a feature of the framework) but no further evidence is provided. I rewrite to make it more clear that this could be an application but should be further investigated as future work (as is done in the conclusion).
A5 Appendix; nodes never deviate by more than one, i.e. which means regular updates. Is this correct? In my understanding this ensures that the total work of all nodes is similar B=20 ensures (somewhat) regular updates
The sampling notation in A5 is a bit weird - but this might be due to my confusion about the role of $tau^i_j$ which makes it hard for me to verify the correctness of the shown procedure.
In the Appendix layer sizes are noted as 4,4,1 - this seems quite small, is there a particular reason for this limitation?
In the Appendix multiple references are broken, where a ? shows instead.
In Section 5 you mention that prior work unifies the objective oriented view. But then you continue with saying that there is no reason to restrict - so what is a unifying or restricting view? I believe this to be mostly a formulation issue.
How would you like to capitalise during the paper, in the main text energy GNN is used but then in the Appendix it is Energy GNN. I would recommend keeping it consistent throughout.
In Section 5, how does the energy function enable robustness? Isn’t this a property of the implicit GNNs and not limited to EGNN?
I did not follow the statement “not necessarily convex with respect to the features”, but the node embeddings, but usually the initial node embeddings are the features? I also think it would be good to make explicit where the convexity is required (in the final convergence proof I assume, following the original proof).
In A.2 there is an additional + on the third line of the proof
The proof of the main insight could be improved for presentation: restating the original theorem, showing the individual assumptions (the second condition where the Ks are set is not immediately obvious), is this why e_i is defined this specific way?
Do you have any intuition on why IGNN is first stable, but then diverges for larger graphs?

---

> ### Author Response · Authors · 2023-11-14
>
> Thank you for your detailed read of our work and helpful comments.
> We have organized our responses below to address what we see as your main concerns, and then answer miscellaneous questions.

---

> ### Author Response · Authors · 2023-11-15
> **Response (1/4)**
>
> **Regarding experiments:**
>
> While inference over large graphs is one application where asynchronous, distributed inference may be desired, there are a number of other applications on small graphs (i.e. multi-agent setting).
> In this work we simply point out large graphs as a possible application (this may have been miscommunicated), but do not focus on this application.
> Instead, our work establishes the conditions under which GNNs become amenable to distributed, asynchronous inference and we propose a novel architecture that meets these conditions.
> We feel it is sufficient to show that our architecture achieves competitive performance on synthetic and benchmark datasets, as the goal is to add to the class of GNNs which are amenable to asynchronous, distributed inference (in the hope that this class of GNNs may be further explored and used in relevant applications).
> Certainly, we hope inference over large graphs is an application that will explored in future work.
>
> The goal of our experiments was to 1) explore a number of synthetic tasks inspired by applying GNNs to parameterize communication between agents in a multi agent system such as a robots swarm (e.g. counting of number of agents, summing agent features, performing localization) where distributed, asynchronous inference would be desired and 2) compare performance of our proposed architecture to existing GNNs on a number of standard benchmark datasets to better gauge its expressive power.
>
> For the synthetic tasks we are explicitly interested in performance during asynchronous, distributed inference; for this reason, we only consider one ‘finite-depth’ or synchronous GNN (a 2 layer GCN) because all GNNs of this flavor will exhibit the same malignancies under asynchronous inference.
> We consider “unreliable-under-asynchrony” GNNs to be a single group and choose one representative, since empirical performance of synchronous architectures under asynchronous execution is not really the focus of the paper.
> We only compare against one other implicit GNN, IGNN, because to our knowledge this is one of the only existing implicit GNN architectures which has convergence guarantees under asynchronous inference.
> EIGNN is another instance that we mention (which we note is extremely similar to IGNN), but we found in our experiments that performance of their architecture depends on solving for the fixed point exactly rather than using an iterative method (see appendix 10); the forward pass thus requires global information, which is not easily amenable to distributed inference.
> Due to global communication requirements of EIGNN’s forward pass and its similarity to IGNN, we exclude comparisons against EIGNN.
> Another architecture we mention is the implicit GNN architecture proposed by scarcelli; however, as we note in the paper, their node update function is not a contraction map over the whole domain and thus does not share the same convergence guarantees as energy GNN and IGNN.
>
> We understand the confusion about emphasis on long-range communication in the experiments; the multi-agent inspired tasks in the synthetic experiment section all essentially require long-range communication for good performance and thus this is discussed in the results section.
> Our goal was not to focus on long range communication, rather we point out that energy GNN is able to accomplish this.
> We see this as an added benefit to energy GNNs (and implicit GNNs in general, as they do not limit the number of rounds of message passing) rather than the main feature.
>
> We would also like to clarify the reported values in Table 4 and 5 in A.10.
> The asterisked values are obtained from other papers.
> We stress that for MUTAG and PROTEINS, these values are likely over-estimates of the test performance, since inspection of the code corresponding to these papers shows that test performance is computed by creating 10 train/test splits, and averaging the *best* test performance achieved during training.
> For our results (not asterisked) we measure mean test performance across 10 splits after a consistent number of epochs that is not determined by test performance for that split, so the value we report are strictly lower than that in the other papers.
> We note that we outperform GCN on MUTAG regardless of if we take the value reported in other papers or the result obtained by our implementation.
> For PROTEINS we outperform GCN when using our evaluation metric and for PPI the accuracy is almost identical.
> As a final note on the discrepancy between energy GNN performance and asterisked IGNN performance on the benchmark datasets: the IGNN architecture from the original paper had multiple layers.
> We do not explore multi-layer energy GNNs in this paper, and leave this to future work.
> It is likely that performance on benchmarks could be improved if multiple optimization layers were used.

---

> ### Author Response · Authors · 2023-11-15
> **Response (2/4)**
>
> **Regarding other strategies to asynchrony:**
>
> There are certainly a number of strategies that one could imagine adopting in making a synchronously-operating GNN robust to asynchronous execution.
> The strategy we explore is architectures which have guaranteed robustness under partial asynchronism by virtue of using either a fixed point iteration or a convex optimization procedure to obtain node embeddings.
> This offers a massive benefit over strategies like training with asynchrony or using the Scarcelli architecture; under mild assumptions, we get *theoretical guarantees* that under asynchronous inference *the same performance can be achieved* as in synchronous inference.
> As we note in the paper, the node update function proposed by Scarcelli is not a contraction map over the whole domain and thus does not share the same convergence guarantees as energy GNN and IGNN.
> We thus exclude comparisons to this work, as it falls into the class of GNNs that are not inherently robust to asynchronous inference.
>
> **Regarding “simplicity” of existing implicit GNNs:**
>
> We do not necessarily claim that existing implicit GNNs are too simple: a more precise statement would be to say they are restrictive and difficult to build on.
> We identify only two existing implicit GNN architectures, IGNN and EIGNN, which are robust to asynchronous inference.
> They have a similar node update function and both parameterize the update function to be contractive.
> Our energy GNN is very different, employing an input-convex GNN, which allows for more flexibility in the the choice of architecture compared to IGNN and EIGNN.
>
> **Regarding efficiency and Figure 2:**
>
> We do not claim that an energy GNN is more efficient than other GNNs.
> As the reviewer points out, during asynchronous inference, it can take thousands of node updates for the dataset loss to converge. (The dataset loss for the chains dataset is the mean binary cross entropy loss between node predictions and targets).
> Note firstly that we used a staleness bound of B=20 in our experiments, which is relatively high.
> Convergence would be faster with a smaller bound.
> Second, we show convergence starting from node embeddings initialized to the zeros vector.
> This initialization is relatively far from the energy minimum, so many iterations are required to converge.
> During what we might call “blank slate” inference, where the node embeddings are not initialized in any informed way, this could be a problem.
> However, in many cases a good initial solution may be available.
> For example, during training we use the solution from the previous epoch.
> During inference in a continuously operating multi agent system, the solution from the previous time step can be used.
> Finally, in Figure 2, we run inference well past the point when 100% dataset accuracy is achieved, in order to show that as the GNN continuous asynchronous updates, the loss remains stable and does not change.
> Good predictions can still be obtained if the process is stopped earlier (e.g. accuracy of 90% corresponds to log loss of approx. -2.2); in fact, an interesting and potentially useful property of energy GNNs and implicit GNNs is that they are an ‘any-time’ algorithm, in which predictions get better with more iterations and a solution can be queried at any point in time.
>
> We also point out that in cases where distributed, asynchronous inference is desired, it may be reasonable to sacrifice efficiency (in terms of number of node updates) since benefits are gained in other places.
>  There is no requirement for central control and coordination of node updates; this removes communication bottlenecks and prevents a single failure point.
> In the application of asynchronous distributed inference over large graphs, memory requirements are distributed and thus reduced per processor.
>
> **Other questions:**
>
> - How long do the implicit GNNs and the EGNN run in practice? How deep are the “infinite depth” gnns in practice?
>     - The number of iterations required to either achieve a fixed point (IGNN) or energy minimum (energy GNNs) will depend on the embedding initialization, the method of solving for the fixed point or the minimum, and the convergence tolerance. Since the number of iterations required for convergence depends on a number of factors, we simply report values from our experiments for reference. During training, we always use the solution from the previous epoch in order to speed up convergence, and our convergence tolerance is 1e-3. For energy GNNs we use L-BFGS to solve for the minimum, typically requiring no more than 10 iterations per graph. For IGNNs we iterate on the fixed point, which may take upwards of 500 iterations.

---

> ### Author Response · Authors · 2023-11-15
> **Response (3/4)**
>
> **Other questions (continued):**
>
> - What is the limiting factor of scaling to larger graphs? Could you go beyond the size of 100 nodes used in the paper, what are the tradeoffs?
>
> We do go to a size beyond 100 nodes in the benchmark datasets; the average number of nodes in a graph in the PPI dataset is **2373.**
> The tradeoff for larger graph sizes is typically seen in the backward pass;
> the size of the linear system being solved to obtain parameter gradients using implicit differentiation is proportional to the number of nodes in the graph.
> - Are the models trained on the same graph sizes that they are run on during inference? Or do you only train on one and then generalize across sizes for the synthetic tasks?
>
> For the counting experiments, we generate n linear graphs of size $1-n$.
> We train on 90% of the graphs and test on 10%; test performance is reported on unseen graphs (with unseen number of nodes). For all other synthetic tasks, the number of nodes is constant across the dataset.
> - The results provided in the main part of the paper are only on individual modes and not reported as the mean over multiple runs if I understand correctly? I see that it might not be straightforward, as you also need to avg. over the used schedule - but I would appreciate it if the results would be averaged over multiple learned models.
>
>  We agree with this feedback, and will run these experiments and report updated values averaged across multiple random seeds for training. Updated results will be up by Friday end of day.
> - The required step size is in O(1/nb) (which seems quite small). Could you provide an intuition on why this has to be dependent on n, to me this is not obvious (I would have thought maybe a graph parameter such as the node degree might be necessary?)
>
> The intuition would be that the dimension of the optimization variable (i.e. the node embeddings for the whole graph) in the overall energy function is proportional to the number of nodes.
> - I was not quite able to follow the task setup for 6.3. I think a precise formulation of the problem using mathematical notation would help to clarify the task and training objective (can also be included in the Appendix).
>
> The task can be summarized as follows: let $D \in \mathbb{R}^{n \times n}$ be the target distance matrix, where $D\_{i,j}= ||Y^i - Y^j||\_2 $ for all $i \in \[1,...n\], j \in \[1,...,n\]$.
> Here, $Y^i \in \mathbb{R}^2$ is the true position of the node, which for random graphs we obtain by uniform random sampling in the unit square, and for lattice graphs we obtain by constructing a triangular lattice graph in the unit square.
> In this task we don’t care about node predictions matching the true positions exactly; instead, we want the nodes to determine their own 2D coordinate system where distances between all nodes in the graph are correct.
> For random graphs, nodes share an edge if their distance is below 0.5, and for lattice graphs, connectivity is determined by the lattice structure.
> Node features are one-hot embeddings of node IDs, and edge features are distances to connected neighbors.
> The loss for each graph in the dataset is $\frac{1}{n^2} \sum\_\{i\=1\}^n \sum\_{j=1}^{n} (||(\hat{Y}\_i - \hat{Y}\_j)||\_2 - D\_{i,j})^2$,
> where $\hat{Y}_i \in \mathbb{R}^2$ is the predicted position of node i, obtained by executing a GNN.
> The loss can be stated in words as the mean squared error between the predicted distance between two nodes, and the actual distance.
>
> - In the related work section, I would encourage to include the work on asynchronous GNNs (rather than the focus on distr. training which is not that close to the presented ideas). This includes the work of Faber et al. and Dudzik et al
>
> We were unaware of this work but agree it is relevant. Faber et. al appear to use an RNN-type architecture per node to generate message outputs and iterate the node state, and attempt to train the architecture to be robust to things like message delays by simulating this process during training. They do not fall into the class of GNNs which is provably robust to asynchronous, distributed inference. Dudzik et. al. point out certain classes of GNNs, namely a specific form of pathGNN that use only max aggregation and use specific message functions (for example, a tropical linear transformation) that depend only on sender nodes, can be executed fully asynchronously. The motivation for the paper is to achieve algorithmic alignment between GNNs and existing asynchronous algorithms, rather than examining asynchronous GNN inference in general.

---

> ### Author Response · Authors · 2023-11-15
> **Response (4/4)**
>
> **Other questions (continued):**
>
> - I have trouble understanding the definition of $\tau_i^j(t)$, shouldn’t it be the last updated timestep rather than the amount by which the embedding is outdated (this in my understanding is the quantity $s\_{ij}$)
>
> We agree the wording is confusing. It would be more clear to say given update times $T^j \subset \\{0,1,2,...\\}$ for node j, $\tau\_i^j(t) \in T^j$ is the time corresponding to node i’s view of node j at time t, where  $s\_{ij} = t -\tau\_i^j(t) \in \[0,t\]$ is the staleness.
>
> - A5 Appendix; nodes never deviate by more than one, i.e. which means regular updates. Is this correct?
>
> Yes, in our experiments the number of updates a node has executed doesn’t deviate from the other nodes in the graph by more than one.
> We thought it was a reasonable assumption that nodes update at some regular frequency, as this depends only on local computation at the node. On the other hand, it is reasonable to assume communication with other nodes is less reliable; we thus set the staleness bound on messages relatively high, being at most outdated by 20 updates.
> An empirical examination of the effect of different levels of staleness and relative frequency of node updates would be interesting, although we can partially predict these effects by observing the relationship between the maximum asynchronous learning rate and staleness bound (as shown in appendix A.4.2).
> Note that the assumptions of partial asynchronous algorithms state that B is both a bound on the amount (in time) that a node can go without updating, and a bound on the message staleness.
> Intuitively, convergence will likely be slower as B increases and the maximum learning rate decreases.
>
> - In Section 5, how does the energy function enable robustness? Isn’t this a property of the implicit GNNs and not limited to EGNN?
>
> Yes, it is not a property limited to energy GNN.
> Energy GNN achieves robustness to asynchrony through optimization of the convex energy function, while implicit GNNs achieve robustness by iterating on a contractive fixed point equation.
>
> - I did not follow the statement “not necessarily convex with respect to the features”, but the node embeddings, but usually the initial node embeddings are the features? I also think it would be good to make explicit where the convexity is required
>
> In our architecture, the node embeddings are defined implicitly as the embeddings which minimize the convex energy function.
> Only convexity with respect to the node embeddings is required for the energy function.
> The node features are provided as fixed inputs to the energy function and there is no requirement that the energy is convex with respect to the node features.
> The solution to the energy minimization does not depend on the initialization of the embeddings (we initialize to zero, or, during training, to the solution from the previous epoch) as the energy is convex (this is the same as for implicit GNNs in general; the same fixed point is reached for the node embeddings regardless of initialization).
>
> - Do you have any intuition on why IGNN is first stable, but then diverges for larger graphs?
>
> We assume you mean why IGNN performance decreases for larger graphs on the chains experiment?
> Generally it is known that contractive updates make long range communication challenging.
> One intuition is provided by Li et al. https://arxiv.org/pdf/1511.05493.pdf, who show that repeated contractive updates decrease the dependence of distant node values.
> The authors of EIGNN argue that they achieve better long range communication than the original IGNN paper by solving for the fixed point directly instead of using iterative solvers.
>
> **Final comments:**
> We have additionally uploaded an updated version of the paper with broken reference links fixed and other concerns addressed (e.g. the description of $\tau_i^j(t) \in T^j$).

---

> > ### Comment · Reviewer_fGVf · 2023-11-17
> >
> > I thank the authors for their detailed feedback and answers to my questions. I appreciate the clarifications, in the following I will focus on the key items that remain from my perspective.
> >
> > Generally speaking, I think centering the paper more around the idea of robustness under asynchrony rather than more efficient at scale / inference is a step into the right direction. In my opinion, this would still require more rewriting of the introduction and current presentation of the work.
> >
> > Evaluating on synthetic tasks is fine in principle, but it should be clear that the comparison is fair and targets the right aspects. For example, in its current form I do not think that the sums and chains tasks perform this adequately. The GCN baseline (which should represent how much you lose if you are not provably robust under the proposed revised storyline) is unable to even solve the task in the synchronous setting due to only executing 2 layers while the task at hand clearly requires more (as it is long range). I would suggest either adapting the GCN baseline or add more baselines which more adequately reflect the performance of synchronous GNNs. I believe that they would still fail under async. execution, but to show that the task would be learnable in principle is very important for a fair comparison. Otherwise, I think the task mainly stays a long range task which would require more baselines which explicitly handle this.
> >
> > I see that a big advantage of Energy GNN is that you get provably robust performance - I would still appreciate an empirical comparison to other works that do not necessarily get this property (Scarselli …). I would be surprised if there is no empirical evidence that the provable robustness performs better (at least on synthetic tasks).
> >
> > Regarding efficiency, I believe the paper would benefit from a consistent metric throughout the whole paper, i.e. number of updates/or number of rounds/or maybe even time measurements - which would make it easier to properly put the performance of the models into the right context. Maybe this metric should take into account B (which essentially is a proxy for how asynchronous the environment is). Moreover, this is essentially the “price” one must pay for achieving robustness - so the tradeoff should be discussed in the main paper.
> >
> > Varia: regarding the size I was mainly wondering about the synthetic tasks if there is a specific reason for the cutoff of 100; regarding A5 I would argue that all nodes execute roughly the same number of updates is the right choice - however, regular updates is enforced through the choice of B rather than the total number of updates;
> >
> > As a last comment, I wonder if you could strengthen your results even more by considering a (learned?) adversarial execution schedule instead of uniform for future work.
> >
> >
> > Overall, I believe this to be a very interesting line of work and if the motivation and evaluation of the proposed method is properly revised I do believe this is worth publishing at a top venue. However, in its current form, I still have too many reservations for recommending acceptance and I will retain my score.

---

> > > ### Author Response · Authors · 2023-11-22
> > >
> > > Thanks for the reply!
> > >
> > > We refer the reviewer to our global comment which states modifications made in the final revised paper, which may address some of your concerns about the focus of the paper.
> > >
> > > Some final comments on your remaining concerns:
> > > - The performance of synchronous GNNs on the synthetic tasks is, in a sense, irrelevant. For each of these synthetic tasks, we assume that inference is constrained to be distributed and asynchronous. If this constraint is not imposed, there is simply no need to even use a GNN, as there are obvious, perfect algorithms for solving each task. For sums, a central controller would simply access all the node features and sum them; the same goes for counting.
> > > - An efficiency metric would definitely be interesting for this work. However, this is actually non-trivial to reason about. For reference, take a look at the lengthy analysis done by “Hogwild!: A Lock-Free Approach to Parallelizing Stochastic Gradient Descent” by Nui et. al. (2011); their analysis relies on simplifying assumptions and still spans several pages. We do not believe such an analysis to be necessary to support the main ideas of the paper, especially since we provide proof of guarantees of convergence during asynchronous inference. Furthermore, a comment by several reviewers was that the paper was already math-heavy. In practice, for asynchronous algorithms such as ours, we believe that these evaluations should probably be done empirically on a per-application basis. This was what we aimed to show in A.5 with plots of dataset loss during asynchronous inference, as a function of number of individual node updates.
> > > - there is no particular reason we cut off the chains experiment at 100 nodes.

---

> > > > ### Comment · Reviewer_fGVf · 2023-11-23
> > > >
> > > > Thank you for the additional clarifications. Regarding the efficiency metric I see that this is not as straightforward and is not absolutely required, nevertheless, I believe if possible it would considerably strengthen the paper.
> > > >
> > > > Regarding the performance of the synchronous GNNs - I strongly disagree with the statement.
> > > > How I understand the experiment for the synthetic tasks is that you would like to show that in the scenario of asynchronous inference "regular" GNNs fail catastrophically while implicit or your proposed Energy GNN can solve the tasks. However, if the task cannot even be solved by the regular GNN in the synchronous setting anyway (because of the 2 layer setting) the comparison seems very unfair to me. As a consequence, the conclusion that the performance drop is due to the asynchrony and not just the GCN model choice is then not well supported, i.e. for the chains the drop is only a few percentages between the regular and asynchronous GCN.
> > > >
> > > > I do understand that the performance of the synchronous GNNs is not the main point of what the experiments try to convey - but taking the aspects I mentioned above into consideration, I still have reservations about the setting of the comparisons. However, I also think that this should be fixable - maybe you could consider graph topologies which have a constant (or same number as layers in the GCN) diameter.

---

### Official Review · Reviewer_zshX · 2023-10-30

**Soundness:** 1 poor
**Presentation:** 1 poor
**Contribution:** 2 fair
**Rating:** 3
**Confidence:** 3

**Summary:**

The paper considers the problem of distributed inference of GNNs. The authors focus on partial asynchronism, which bounds the time between updates across each node and the the amount of data that can be outdated at each node. They then present how GNN inference is performed under the partial asynchronism. Finally they propose an architecture, called Energy GNN in which node embeddings are computed by minimizing an energy function which is amenable to partially asynchronous inference.

**Strengths:**

The problem of distributed executions in the context of GNNs is interesting.

**Weaknesses:**

I think the paper only considers distributed inference, but does not deal with distributed training. I am not sure I understand why one would want only the inference to be distributed if the training is not. More precisely, if the graphs are so large that inference needs to be distributed, how was the model even trained?

The paper is hard to follow and tends to lack clarity. The appendix is confusing and it contains broken pointers (see the number of "??") thus I did not find it useful in understanding the paper better. Also what is the meaning of the "*" in the tables?

The experimental section is limited and tends to be not very clear. I think it should explore much larger graphs having millions of nodes, and certainly not those having 100 nodes that can be easily ran by standard non-distributed GNNs. For example, what about the ogb MAG240M dataset?

**Questions:**

1. Please evaluate on large scale datasets, e.g., MAG240M, where distributed executions are needed.
2. Why some entries of the tables have the std while some others have not?

---

> ### Author Response · Authors · 2023-11-15
>
> Thank you for your time in reviewing our work. We believe there may have been a misunderstanding about the focus of the paper. While inference over large graphs is one application where asynchronous, distributed inference may be desired, there are a number of other applications on small graphs (i.e. multi-agent setting). In this work we point out large graphs as a possible application, but do not focus on this per se. Our synthetic experiments are particularly motivated by tasks that would be relevant to multi-agent systems (e.g. counting of number of agents, summing agent features, performing localization). In these applications, asynchronous and distributed inference is absolutely desirable, as it has the benefits of not requiring central control, which introduces algorithmic and communication overhead as well as a central point of failure for the system.
>
> Furthermore, it is entirely possible that distributed inference might be desirable even if the GNN was trained in a non-distributed manner. Training and inference can have vastly different resource requirements.  Models are only trained once and so that can be done in a centralized way if desired, but inference will be done indefinitely and is constrained by whatever device(s) the network will be deployed to. To use a modern example of this discrepancy: although it cost tens of millions of dollars for OpenAI to train ChatGPT, that cost is exceeded by a single week of providing inference as a service with the model. Finally, distributed inference and distributed training are essentially independent problems. For instance, for extremely large graphs we could have trained using an existing distributed training method.
>
> We also point out that while the maximum graph size in our synthetic experiments is 100 nodes, we also run on the PPI dataset, where the average graph contains over 2000 nodes.
>
> **Responses to other questions:**
>
> - Why some entries of the tables have the std while some others have not?
>
> A standard deviation is always included for running GCN asynchronously, since under asynchronous execution the predictions vary across random runs. We perform 10 random runs to obtain these standard deviations. For IGNN and energy GNN, performance does not depend on the run since these architectures guarantee convergence of embeddings under asynchronous execution, so no standard deviation is provided. For the 10 node counting experiment, the dataset is extremely small and consists of only 10 graphs, so we report average performance across 10 folds of the data. Finally, for the MUTAG and PROTEINS datasets, we report mean accuracy and standard deviation across 10 splits of the dataset. For all other table entries where standard deviation was not included, it is either because we report performance on a single test set from the dataset, or because the architecture gives the same predictions under asynchronous execution.
>
> That being said, we have decided for all experiments to run across 10 folds, as well as with random network parameter initializations. After this, standard deviations will be included in all table entries. We will have updated results uploaded to the paper by end of day Friday.
> - The paper is hard to follow and tends to lack clarity. The appendix is confusing and it contains broken pointers (see the number of "??") thus I did not find it useful in understanding the paper better.
>
> Could you expand on what lacked clarity or was hard to follow? We have fixed the broken pointers in the appendix and uploaded an updated version of the paper.
> - Also what is the meaning of the "*" in the tables?
>
> The asterisked values are obtained from other papers. We stress that for MUTAG and PROTEINS, these values are likely over-estimates of the test performance, since inspection of the code corresponding to these papers shows that test performance is computed by creating 10 train/test splits, and averaging the *best* test performance achieved during training. For our results (not asterisked) we measure mean test performance across 10 splits after a consistent number of epochs that is not determined by test performance for that split, so the value we report are strictly lower than that in the other papers. We note that we outperform GCN on MUTAG regardless of if we take the value reported in other papers or the result obtained by our implementation. For PROTEINS we outperform GCN when using our evaluation metric and for PPI the accuracy is almost identical.

---

> > ### Comment · Reviewer_zshX · 2023-11-22
> >
> > I would like to thank the authors for the time spent answering my questions.
> >
> > Despite the answers, I believe that the work requires a few more iterations before being presented at a conference. In particular, I would recommend the authors to improve the presentation making the focus of the paper clearer and, importantly, aligned to the experimental evaluation. I think that if inference over large graphs is presented as an application throughout the paper, then the experimental evaluation should support the statement.
> >
> > I would recommend expanding the experimental evaluation: currently there are mainly synthetic datasets and limited baselines. The PPI dataset the authors are referring to is only one, and it is also not particularly large.
> >
> > Finally, I would like to clarify that what the authors are describing for MUTAG and PROTEINS is the standard evaluation procedure [1]. I would encourage the authors therefore to use exactly the same spilts and evaluation procedure as all the other baselines (and further include the other additional TUDatasets [2] instead of choosing these two only).
> >
> >
> > [1] Xu et al. 2019. How Powerful are Graph Neural Networks? ICLR 2019
> > [2] Morris et al. 2020. TUDataset: A collection of benchmark datasets for learning with graphs. ICLM GRL+ 2020

---

> > > ### Author Response · Authors · 2023-11-22
> > >
> > > Thank for reading through our reply, and for the response.
> > >
> > > We've uploaded a revised version of the paper where we have de-emphasized the large graph application and instead focused on the synthetic tasks motivated by multi-agent systems, to be more in line with the experiments we present. While inference over large graphs remains a glaring example of a relevant application for asynchronous GNNs, we believe this exploration should be left to future work (particularly since this would require engineering a distributed training pipeline that is amenable to energy GNNs, where a linear system depending on all node embeddings has to be solved in the backward pass to obtain gradients).
> > >
> > > In our revised paper, when we introduce the experiments we explain our choice for comparisons against other models, so it is more clear why comparisons are limited. We emphasize again that our non-synthetic experiments are limited because we focus on tasks which require distributed, asynchronous inference (since this is the main focus of the paper); the benchmark tasks don't really belong to this group. This is also why we report performance on benchmarks in the appendix rather than in the main paper.
> > >
> > > Regarding the evaluation process for MUTAG and PROTEINS; we are aware of the evaluation strategy of [1], and point out that the values reported for IGNN do not follow this strategy. Rather than taking the mean performance across all folds for each epoch, and reporting the max mean performance, the authors of IGNN appear to first take the max performance for each fold, then report the mean of the max performance.

---

### Official Review · Reviewer_Ea9C · 2023-10-31

**Soundness:** 2 fair
**Presentation:** 3 good
**Contribution:** 2 fair
**Rating:** 5
**Confidence:** 3

**Summary:**

One major issue in running graph neural networks (GNN), at inference time, in a distributed fashion is that GNNs require synchronous communication between layers of the graph, but distributed execution is asynchronous which means communications between nodes in a GNN are done at different times or can have stale communications. This can lead to the GNN inference failing. However, some types of GNNs can handle asynchronous execution which are termed “Hogwild” GNNs. This paper introduces a ‘hogwild’ GNN architecture termed “energy GNN” which views message passing (communication) between nodes as a convex optimization problem during training and, as a result, can run asynchronously during inference, in addition to having good performance compared to modern GNN architectures.

**Strengths:**

•	Good introduction that clearly explains the problem and good grammar throughout the paper.
•	Comprehensive experimental evaluation done on multiple datasets.

**Weaknesses:**

•	Some GNN related terminology is difficult to follow, due to not having explicit definitions
•	Section 4 last paragraph difficult to understand.
•	Paper is math-heavy in several sections.
•	Many references to appendix, which is not visible.

**Questions:**

One issue in this paper is that it uses terminology which is not explicitly defined (if such terminology is commonly known in the target audience, this may not be big issue). For example, Finite depth GNN, contractive (in context of GNNs), and energy function have no explicit definition. Implicit GNN and fixed point GNN seem to be used interchangeably. IGNN is first used in the evaluation section without being defined, (is it referring to Implicit GNNs?).

In Section 4, the last paragraph which explains the advantages of a fixed-point GNN over finite depth GNNs, it is unclear what the benefit is, due to insufficient explanation. For example, what do “change its output in response to changes in the inputs” and “coordinating another forward pass of the network” mean, and why is it important for GNNs?

The experiments section is easy to follow, since each experiment and its results are explained clearly. In addition, several different datasets used for experiments, which makes the benefits of the proposed architecture stand out. However, only a total of 3 GNN architectures are tested. Having more architectures or explaining why the chosen architectures are sufficient would be good to include. Furthermore, in all experiments we see the proposed energy GNN performs best, but is this always the case and could there be cases where energy GNN fails to perform well?

Lastly, the paper is also math-heavy in several sections, which makes it difficult to follow for readers unfamiliar with GNNs. Perhaps some figures could be used to supplement the math sections.

---

> ### Author Response · Authors · 2023-11-15
> **Response part 1/2**
>
> Thank you for your time taken to review our paper. We address your concerns and questions in this comment and the following one.
>
> We agree that several of the terms we used could have been clearer, and may have lead to confusion. Regarding each of the terms you mention:
>
> - Finite depth GNNs are GNNs that have a fixed number of layers of message passing. Infinite depth GNNs are defined in the background section.
> - We use the term “contractive” short for “contraction map” throughout the paper as an attribute of functions (e.g., “update functions” in the abstract, “message passing function” in 2.3, etc.). A function $f$ is a ***************contraction map*************** with respect to its argument, which is to say $\exists \mu \in [0, 1)$ such that $||f(x) - f(y)|| \leq \mu ||x - y||$ for any $x, y$ in the domain of $f$. Informally this means that the distance between any two points *********decreases********* after we apply the function. In the context of GNNs, the argument in question is the node embedding (which we denote with $h$ in this work).
> - We use the term “energy function” to refer to the objective function associated with the optimization problem. In the first two paragraphs of section 5 we explain the optimization-based view of GNN node updates and proceed to call the objective function being optimized an ‘energy function’
> - We do use implicit GNN and fixed point GNN somewhat interchangeably, which is not technically accurate. Implicit GNNs (as a concept) refer to GNNs in which the mapping from data (features) to node embeddings is given implicitly rather than explicitly. Optimization based GNNs and fixed point GNNs are both implicit GNNs in this broad view. In our evaluation section, we are overloading “IGNN” to mean the specific architecture presented in the paper “Implicit GNNs” Gu et al. 2020.
>
> We have made several edits to the main paper to attempt to clarify this terminology.
>
> Regarding the final paragraph of section 4, it may be clarifying to provide an example. Suppose the input graph has dynamic/time-varying node or edge features; for instance, a sensor network where where the node features correspond with sensor measurements (which change over time). Consider the case of using a finite-depth GNN to generate predictions. At some time $t$, a forward pass is executed, in which each node performs a fixed number of rounds $L$ of message passing to generate predictions. If at some time $t’ > t$ the features of the nodes or edges change, the node predictions will now be outdated. To update predictions, another forward pass of the network needs to be coordinated. Coordinating with a global controller is non-trivial, fragile, and adds communication overhead and bottlenecks. Distributed coordination amongst the nodes is similarly non-trivial. The implicit GNN operates continuously, repeatedly updating its embedding, and changes in feature inputs are handled ‘for free’ in the sense that the network does not need to re-start a forward pass. This feature is not necessarily important for GNNs in general, but it certainly is for applications where we want distributed GNN inference and the graph features are dynamic inputs. We do not consider any applications where this is relevant but hope to pursue this in future work.

---

> ### Author Response · Authors · 2023-11-15
> **Response part 2/2**
>
> We understand that the motivation behind our choice of architectures to compare to in experiments may not have been clear in the paper.
>
> - For the synthetic tasks we are concerned with performance during asynchronous, distributed inference. We only consider one ‘finite-depth’ or synchronous GNN (a 2 layer GCN) because all GNNs of this flavor will exhibit the same malignancies under asynchronous inference. We compare against IGNN, because to our knowledge this is one of the only existing implicit GNN architectures which converges under asynchronous inference. EIGNN is another instance that we mention (which we note is extremely similar to IGNN), but we found in our experiments that performance of their architecture depends on solving for the fixed point exactly rather than using an iterative method (see appendix 10); the forward pass thus requires global information, which is not easily amenable to distributed inference. Due to global communication requirements of EIGNN’s forward pass and its similarity to IGNN, we exclude comparisons against EIGNN. Another architecture we mention is the implicit GNN architecture proposed by Scarcelli; however, as we note in the paper, their node update function is not a contraction map over the whole domain and thus does not share the same convergence guarantees as EGNN and IGNN.
> - For the real world experiments we only compare against our implementation of GCN and IGNN; however, we do provide values reported in other papers for other architectures. We asterisk these values in our tables because their evaluation metrics are biased to overstate the typical-case performance. Inspection of the code corresponding to these papers shows that test performance is computed by creating 10 train/test splits, and averaging the *best* test performance achieved during training. For our results (not asterisked) we measure mean test performance across 10 splits after a consistent number of epochs that is not determined by test performance for that split, so the value we report are strictly lower than that in the other papers. We could have added a more comprehensive comparison, but as our goal was not to show SOTA performance, we believe the context we provide for performance of other methods was sufficient.
>
> Other questions:
>
> - Furthermore, in all experiments we see the proposed energy GNN performs best, but is this always the case and could there be cases where energy GNN fails to perform well?
>     - Certainly there are cases where energy GNNs will fail to perform ‘well’, in the sense that other GNN architectures may perform better (this is certainly the case for our real-world dataset experiments; while we achieve competitive performance with existing architectures, we do not achieve SOTA performance). However, most architectures are not robust to asynchronous inference; energy GNNs are. What we attempt to show in our experiments is that energy GNNs both have this property and are reasonably performant.

---

> > ### Comment · Reviewer_Ea9C · 2023-11-22
> >
> > While I appreciate the time taken by the authors to answer my concerns, it seems to me that the paper would benefit from one round of revision.

---

### Official Review · Reviewer_e2vg · 2023-11-03

**Soundness:** 3 good
**Presentation:** 3 good
**Contribution:** 3 good
**Rating:** 6
**Confidence:** 2

**Summary:**

This paper develops energy GNN, a novel architecture for distributed GNN inference with asynchronous communication which can be applied to robotics, remote sensing, and other domains. Most existing GNN architecture cannot handle this problem. Energy GNN leverages input-convex GNN to ensure convergence.

**Strengths:**

- This paper explains the related works and background in details, and the proposed method is well-motivated.
- The research problem studied in this has some practical applications but was seldom studied before.
- The proposed algorithm is interesting and intriguing.
- Experimental evaluation shows that Energy GNN works well on the synthetic datasets.

**Weaknesses:**

- The experiments are not sufficient.

  - The convergence curves of Energy GNN are not reported.

  - The efficiency and time complexity of Energy GNN are not reported.

- The accuracy performance is relatively worse than GCN on real-world datasets (e.g., PPI and MUTAG).

**Questions:**

- What's the performance of GCN (async) on real-world datasets?

- Is $\tau_i^j(t)$ in Equation 8 a staleness bound? If yes, how to set this term in your evaluation? What's the effect of changing this term?

- It would be great if the authors can provide a similar histogram in Figure 1(d) for Energy GNN.

- Some reference links in the supplementary material are broken.

- The citation format is not consistent. Some first names are abbreviated (e.g., A Bojanczyk) but some are not.

---

> ### Author Response · Authors · 2023-11-15
> **Response (1/2)**
>
> Thank you for taking the time to read and comment on our paper. We first address your main concerns then answer the specific questions posed.
>
> Regarding your concerns about providing convergence curves for energy GNN: do you mean at inference time, or at training time? We provide curves demonstrating the convergence of the log dataset loss during asynchronous energy GNN inference for the chains experiment (in Appendix A.7, Figure 2). While we can certainly provide these curves for other experiments, the purpose of Figure 2 is to illustrate empirically that energy GNNs (and IGNNs) are robust to asynchronous inference, supplementing the theoretical guarantees for convergence that we show hold in appendix A.4. If the reviewer instead means that training curves are not provided, we can certainly provide those.
>
> Regarding concerns about providing efficiency and time complexity for energy GNN, we again would like to clarify 1) what is meant by efficiency and 2) if the reviewer means during inference time or during training.
>
> - For time complexity at training time, we have two considerations: the number of iterations required to minimize the energy with respect to the embeddings, and the time required to solve the linear system during implicit differentiation in the backward pass. The time complexity for minimizing the energy depends on the method used to minimize it (we can use any convex optimization procedure). We use iterative solvers for the forward pass (specifically LBFGS) so the actual number of iterations required for convergence depends on where the embeddings are initialized. In practice, we always initialize the embeddings to the energy-minimizing embeddings from the previous epoch of training, which makes convergence fast since small changes in energy parameters (like those from one epoch to the next) typically correspond to small changes in the energy minimizing embeddings. The time complexity for the backward pass also depends on the method used to solve the linear system induced by implicit differentiation, which is a sparse linear system that can be solved efficiently with a method like conjugate gradient. The same approach of initializing the solution to be the solution from the previous epoch makes the backward pass converge faster.
> - During asynchronous inference, the time complexity is much more difficult to reason about and is not trivial to derive, requiring an analysis akin to that done in “Hogwild!: A Lock-Free Approach to Parallelizing Stochastic Gradient Descent” by Nui et. al. (2011). We do not believe this analysis to be necessary to support the main ideas of the paper.
>
> Regarding concerns about performance on real-world datasets, we first would like to clarify the reported values in Tables 4 and 5 in A.10. The asterisked values are obtained from other papers. We stress that these values are likely over-estimates of the test performance, since inspection of the code corresponding to these papers shows that test performance is computed by creating 10 train/test splits, and averaging the *best* test performance achieved during training. For our results (not asterisked) we measure mean test performance across 10 splits after a consistent number of epochs that is not determined by test performance for that split, so the value we report are strictly lower than that in the other papers. We note that we outperform GCN on MUTAG regardless of if we take the value reported in other papers or the result obtained by our implementation. For PROTEINS we outperform GCN when using our evaluation metric and for PPI the accuracy is almost identical.
>
> We also emphasize that our goal is not necessarily to outperform other GNNs on either the real world or synthetic experiments. Our overarching goal is instead to provide a novel architecture for an asynchronous-friendly GNN (of which there are very few, IGNNs being the prime example that we compare against) and evaluate its representational power. We do not claim that this is a superior GNN architecture for all use cases, but for situations where asynchronous, distributed inference is required, we believe we have shown that energy GNNs are a competitive architecture. Even compared to GNNs operating in the synchronous regime (in our real-world experiments) we achieve competitive performance.

---

> ### Author Response · Authors · 2023-11-15
> **Response (2/2)**
>
> **Responses to questions:**
>
> - What's the performance of GCN (async) on real-world datasets?
>
> GCN (and other architectures assuming synchrony across layers) does not provide reliable predictions under asynchronous execution. If reliable predictions are desired, these GNNs are automatically disqualified. When we perform experiments with real-world datasets, we momentarily ignore asynchronous inference, and instead provide an initial evaluation of how expressive/powerful energy GNNs are relative to other GNNs on a level (synchronous) playing field.
>
> - Is $\tau\_i^j(t)$  in Equation 8 a staleness bound? If yes, how to set this term in your evaluation? What's the effect of changing this term?
>
> $\tau\_i^j(t)$  is not a staleness bound, it is the time step corresponding to node i’s view of node j at time t. This time step may correspond to any time in [0, t]; under partial asynchrony, we assume a staleness bound B, where $t - B < \tau_i^j(t) \leq t.$ This information is found in appendix A.1 which discusses partial asynchrony; if there are confusions or there is some information that should be in the main paper, please let us know! We set the staleness bound to 20 in our asynchronous inference experiments; the maximum step size that can be used for energy minimization is inversely proportional to $B$ (see appendix A.4), so convergence will tend to be slower the more staleness is introduced.
> - It would be great if the authors can provide a similar histogram in Figure 1(d) for Energy GNN.
>
> Energy GNNs, given enough iterations and small enough step size, always converge to the same value (within some numerical tolerance). The histogram in this case would just be a single peak (again, within some numerical tolerance).
>
> - Some reference links in the supplementary material are broken / The citation format is not consistent. Some first names are abbreviated (e.g., A Bojanczyk) but some are not.
>
> Fixed in new upload, thank you for pointing this out.

---

### Author Response · Authors · 2023-11-22
**Comment on revisions in final paper upload.**

Thank you to all the reviewers for their time.

As a final response to all reviewers, we’d like to describe the changes made in the revised submission.
- There appeared to be a lack of clarity on the goal of the paper, with several reviewers saying that inference on large graphs was the focus of our work, as presented. We have overall de-emphasized the large graph application and instead focused on the synthetic tasks motivated by decentralized multi-agent systems, to be more in line with the experiments we present. Changes include removing the sentence on large graph inference in the abstract, reordering parts of the introduction, and re-writing the conclusion.
- We have explained our choice of architectures to compare against in the Experiments section to address why more ‘finite-layer’, conventional GNNs are not considered.
- We have added results for experiments with different random seeds, per the reviewers’ request.
- We have removed emphasis on long-range communication in describing synthetic tasks as this is not their focus (the focus is on tasks relevant to multi-agent distributed systems).
- all inconsistent format citations and broken references are fixed.

---

### Meta-Review · Area_Chair_wHTu · 2023-12-12

**Metareview:**

Summary: The work considers distributed inference in GNN, reframing message passing as an optimization procedure to achieve robustness to asynchrony.

Strengths: Generally, the referees agree the considered problem and the execution model novel and interesting. Referees find the paper mostly well written and provides good description of related works, except for some complaints in the appendix.

Weaknesses: Referees find the experimental evaluation was partly comprehensive but missed some metrics. One referee indicates that distributed training is an important component that is not covered and that experiments should include evaluation of (much) larger graphs where distributed executions might be more relevant. Authors point out that training and execution might have different requirements and distributed execution is not only interesting in case of large graphs, but acknowledge relevance of large graphs which they however defer to future work. Another referee also finds that centering the paper around robustness rather than efficient scale is a step in the right direction, but that this will require rewriting and fair comparison in the experiments, concluding with some reservations about the setting of the comparisons.

At the end of the discussion, there is is a positive recommendation with low confidence and two confident recommendations indicating the paper is not good enough. The reviewers conclude that despite the answers, the work requires a revision to better align focus and experiments as well as include further experiments. I conclude the work is very promising but not ready and hence I must reject it.

**Justification For Why Not Higher Score:**

Weaknesses outweigh strengths.

**Justification For Why Not Lower Score:**

NA

---

### Decision · Program_Chairs · 2024-01-16

Reject